# Locally emitted fungal spores serve as high-temperature ice nucleating particles in the European sub-Arctic

Jürgen Gratzl[1], Alexander Böhmländer[2], Sanna Pätsi[3], Clara-E. Pogner[4], Markus Gorfer[4], David Brus[5], Konstantinos Matthaios Doulgeris[5], Florian Wieland[1], Eija Asmi[5], Annika Saarto[3], Ottmar Möhler[2], Dominik Stolzenburg[1], and Hinrich Grothe[1]

[1]Institute of Materials Chemistry, TU Wien, Vienna, 1060, Austria
[2]Institute of Meteorology and Climate Research, Atmospheric Aerosol Research (IMK-AAF), Karlsruhe Institute of Technology (KIT), Karlsruhe, 76121, Germany
[3]Biodiversity Unit, University of Turku, Turku, Finland
[4]Center for Health and Bioresources, AIT Austrian Institute of Technology GmbH, Tulln, 3430, Austria
[5]Finnish Meteorological Institute, Atmospheric Composition Research, Helsinki, Fl-00101, Finland

**Correspondence:** Hinrich Grothe (hinrich.grothe@tuwien.ac.at)

**Abstract.**

Primary biological aerosol particles (PBAPs) can influence weather and climate by acting as high-temperature ice nucleating particles (INPs), especially in clean, rural regions like the European sub-Arctic. However, the actual contribution to atmospheric ice nucleation and exact identity of PBAPs serving as INPs remains poorly understood.

Here, we present measurements of INPs and highly fluorescent aerosol particles (HFAPs) over the course of one year, at the Pallas Atmosphere-Ecosystem Supersite in the Finnish sub-Arctic, aiming to determine whether PBAPs significantly contribute to atmospheric INPs and to identify which types do so. Our findings indicate that certain HFAPs are strongly influenced by meteorological variables, with high concentrations occurring when the station is within the atmospheric mixing layer, suggesting local biological sources. These HFAPs are the main contributors to high-temperature INPs, with an exceptionally

strong correlation (r = 0.94, p < 0.0001) between HFAP concentrations and INPs active at –13.5 °C. For the first time, to the best of our knowledge, we combine INP and HFAP data with direct fungal spore counts and environmental DNA (eDNA) analysis to determine the biological origins of HFAPs and INPs. The results suggest that most high-temperature INPs are likely fungal spores. eDNA analysis further reveals that airborne fungi are dominated by Basidiomycota and that only a small fraction of the detected fungal genera has, to date, been tested for ice nucleation activity (INA) according to the literature.

Among those reported in the literature, most exhibit very low or no INA. This underscores the significant knowledge gap in our understanding of biological ice nucleation in the atmosphere.

## 1   Introduction

Atmospheric aerosols are important and highly variable constituents of the atmosphere. They impact weather and climate by directly scattering or absorbing solar radiation or by changing the radiative properties and lifetime of clouds by acting as cloud

condensation nuclei (CCN) or ice nucleating particles (INPs) (Masson-Delmotte et al., 2021; Pruppacher et al., 1998). The

latter constitute only a minor fraction of the total aerosol population (Vogel et al., 2024; Burrows et al., 2022; Ladino et al., 2019; Si et al., 2018; Boose et al., 2016) and measurements of INP concentrations remain scarce. Measuring and modeling INPs is challenging, due to their high spatial and temporal variability and the complexity of their sources (Herbert et al., 2025; Burrows et al., 2022; Welti et al., 2018). Observations of INPs are particularly rare in Arctic and sub-Arctic environments, where the temperature increase due to anthropogenic climate change is 1.5 to 4 times faster than the global average (Serreze and Barry, 2011; Cohen et al., 2014; Rantanen et al., 2022), a phenomenon known as Arctic amplification. Although the ice or snow albedo effect is considered the primary driver of Arctic amplification (Screen and Simmonds, 2010), cloud feedbacks are the major uncertainties in climate projection models (Ceppi et al., 2017; Zelinka et al., 2020). Model studies suggest that the radiative properties of low-level Arctic clouds are sensitive to variations in INP concentrations (Gjelsvik et al., 2025; Xie et al., 2013). This indicates that the abundance of INPs and the temperatures at which they become active may be a crucial factor in understanding Arctic amplification (Murray et al., 2021; Tan et al., 2022). In particular, the imprecise representation of biological INPs can significantly impact the overall accuracy of INP predictions in climate models (Cornwell et al., 2023).

INPs active at temperatures below -15 °C are generally dominated by mineral dust (Irish et al., 2019; Kanji et al., 2017; Murray et al., 2012; DeMott et al., 2003). Although biological ice nucleation was observed earlier, it was in the 1970s that bacterial cells were systematically shown to nucleate ice at relatively high sub-zero temperatures (Maki et al., 1974; Schnell and Vali, 1976; Maki and Willoughby, 1978). However, atmospheric primary biological aerosol particles (PBAPs), like fungal spores, pollen and bacteria have historically received little attention from atmospheric scientists (Kabir et al., 2020; Xie et al., 2020; Fröhlich-Nowoisky et al., 2016). More recently, emerging evidence suggests, that the majority of atmospheric INPs active above -15 °C are of biological origin, as indicated by heat sensitivity tests (e.g. Hartmann et al. (2021); Šantl Temkiv et al. (2019); O'Sullivan et al. (2018); Garcia et al. (2012); Christner et al. (2008)) and single particle fluorescence measurements (Taketani et al., 2025; Kawana et al., 2024; Pereira Freitas et al., 2023; Cornwell et al., 2023; Schneider et al., 2021; Twohy et al., 2016; Mason et al., 2015; Wright et al., 2014; Tobo et al., 2013). Both techniques are valuable, but neither alone is sufficient to pinpoint the exact types of bio-INPs. Additionally, heat tests may deactivate non-biological INPs, introducing uncertainties (Daily et al., 2022). Single particle fluorescence spectrometers such as the Wideband Integrated Bioaerosols Sensor (WIBS) used in this study, currently lack the ability to reliably differentiate PBAPs without additional instrumentation. For example, a wide variety of biological, natural and anthropogenic aerosols have been associated with overlapping fluorescence channels of the WIBS. An overview of fluorescent particles detected by the WIBS in field and laboratory studies can be found in Gratzl et al. (2025).

To investigate what types of PBAPs exhibit ice nucleation activity (INA), several laboratory studies have been conducted. INA has been identified in pollen from phylogenetically diverse plants (Pummer et al., 2012; Diehl et al., 2001) and pollen constituents (Augustin et al., 2013; Pummer et al., 2015; Burkart et al., 2021; Matthews et al., 2023; Wieland et al., 2025). Additionally, numerous strains of ice nucleation active bacteria have been isolated from clouds or precipitation (Failor et al., 2017; Šantl Temkiv et al., 2015; Joly et al., 2013; Stephanie and Waturangi, 2011).

In contrast, less is known about the abundance and diversity of atmospheric ice nucleation active fungal spores. To date, only a small fraction of fungal species or genera have been tested for INA, and relatively few have been shown to nucleate ice at

temperatures above -15 °C (Tarn et al., 2025; Haga et al., 2014, 2013; Morris et al., 2013; Pummer et al., 2013; Huffman et al., 2013; Pouleur et al., 1992; Jayaweera and Flanagan, 1982). Ascomycota (sac fungi) and Basidiomycota (club fungi) are the two major Phyla of fungi. Recent mechanistic insights by Schwidetzky et al. (2023) have demonstrated that INPs of the fungus *Fusarium* (Ascomycota) consist of small ($\approx 5.3$ kDa) protein subunits that assemble into larger complexes. These findings highlight that fungi can produce highly efficient biological INPs, although their prevalence in the atmosphere remains poorly understood. Among those fungi tested for INA, most belong to the Ascomycota. However, a high abundance and diversity of Basidiomycota is expected in the atmosphere (Niu et al., 2024; Maki et al., 2023; Qu et al., 2021; Huffman et al., 2013). In the boreal forest which spans large areas of the sub-Arctic, Basidiomycota could dominate over Ascomycota (Qu et al., 2021; Sterkenburg et al., 2015). Nevertheless, Sanchez-Marroquin et al. (2021) found that most spores associated with INPs collected during aircraft campaigns over the southeast UK, resembled Ascomycota. Atmospheric fungal composition likely varies by region and ecosystem type, and findings from mid-latitude regions may not directly reflect the atmospheric fungal composition in boreal or sub-Arctic regions. Given the vast diversity of fungal species many of which remain uncharacterized for INA, fungal spores, particularly from forest ecosystems, may represent a significant and underexplored source of atmospheric INPs. Recent studies have identified terrestrial environments as important, though not exclusive, sources of biological high-temperature INPs (active above –15 °C) in the Arctic (Jensen et al., 2025; Wieber et al., 2025; Tobo et al., 2024; Pereira Freitas et al., 2023; Conen et al.). In northern latitudes, high-temperature INPs show a distinct seasonal pattern, with concentrations peaking in summer during snow- and ice-free periods (Barry et al., 2025; Tobo et al., 2024; Pereira Freitas et al., 2023; Schneider et al., 2021; Wex et al., 2019). Fu et al. (2013) and Pereira Freitas et al. (2023) observed similar seasonal cycles in arabitol and mannitol, chemical tracers of fungal spores (Bauer et al., 2008), which they also attributed to terrestrial sources, in contrast to Arctic haze which dominates the aerosol composition during winter and early spring (Beck et al., 2024; Asmi et al., 2011; Quinn et al., 2007).

These findings imply that high-temperature INPs may become more prevalent in a warming climate, due to arctic greening and prolonged snow free periods in the northern boreal forest (Barry et al., 2025; Berner et al., 2020; Tobo et al.; Dankers and Christensen, 2005) and that fungal spores might contribute significantly to the highly active INP population. Since INPs influence cloud microphysics, particularly the phase, lifetime, and radiative properties of mixed-phase clouds (Bellouin et al.), changes in their abundance or composition can have broader implications for Arctic cloud feedbacks and climate sensitivity. More data on the concentration and biological origin of high-temperature INPs in northern latitudes could therefore enhance our understanding of cloud feedback mechanisms and potentially improve present and future climate predictions. Given the vast extent of the boreal forest, it should be considered a potentially critical source of biological INPs. In particular, fungal spores, which are abundant in boreal ecosystems, may represent a key fraction of these biologically derived INPs. Building on this, we hypothesize that locally emitted fungal spores from the northern boreal forest are a major source of high-temperature INPs in the European sub-Arctic. We present measurements of INPs and PBAPs from the Pallas Atmosphere-Ecosystem Supersite. We combine single particle fluorescence measurements with direct fungal spore counts and complement these data with eDNA sequencing of airborne fungi to elucidate the biological origin of-high temperature INPs in this region.

## 2 Methods

### 2.1 Observation site

Measurements were conducted at Sammaltunturi station at the Pallas Atmosphere-Ecosystem Supersite in Finnish Lapland within the Pallas-Yllästunturi-National Park. Pallas is located 170 km north of the Arctic Circle, at the northern edge of the boreal forest, and experiences a sub-Arctic climate (Köppen, 1931; Beck et al., 2018). Sammaltunturi station (67.9733° N, 24.1157° E) is situated on top of Sammaltunturi hill at an elevation of 565 m above sea level, approximately 100 m above the treeline. As a result, vegetation in the immediate vicinity consists primarily of low vascular plants and lichen (Hatakka et al., 2003). The air at Pallas is considered representative of the clean continental background air of northern Europe, as there are minimal local or regional air pollution sources (Doulgeris et al., 2022; Lohila et al., 2015).

### 2.2 Setup and timeline

The campaign took place in two parts: The first part was integrated into the Pallas Cloud Experiment 2022 (PaCE22) campaign from September to December 2022 and the second part of the campaign was from April to September 2023. The instruments were operated at either one or both parts of the campaign. The times when each instrument was operated is shown in Fig.1 (b). Additionally to the instruments described in Sect. 2.3 to 2.8, an Aerodynamic Particle Sizer APS 3321 (TSI, USA, aerodynamic diameter 0.5–20 µm) was operated at the station and was used to confirm meaningful data collection by the WIBS (see Supplement Sect. S1). The setup can be seen in Fig.1 (a). The Numbers beside the instruments refer to the flow rate ($L \min^{-1}$). The main inlet for aerosol instruments at the Sammaltunturi station is a whole air inlet with no cut-off diameter and thus collecting aerosols effectively also during fog and in-cloud periods. The inlet is ACTRIS (Aerosol, Clouds and Trace Gases Research Infrastructure) approved, and is located 2 m above the station's roof and approximately 6 m above ground. The inlet is slightly heated (to about 1-2 °C) to avoid snow and ice accumulation outside when the station is inside clouds or when it is snowing. The sampling into the aerosol instruments is conducted at room temperature. The aerosol instruments connected to the inlet have separate Permapure MD-700 nafion dryers for sample drying. More details about the inlet and sampling can be found in Backman et al. (2025) and Komppula et al. (2005). All concentrations, except for the Burkard trap are reported for standard conditions (101325 Pa, 273.15 K). If not explicitly written otherwise, all data is reported in Eastern European Time (UTC+2h), which is used as station time in Sammaltunturi.

### 2.3 Single particle fluorescence measurement

A Wideband Integrated Bioaerosol Sensor 5/NEO (WIBS, Droplet Measurement Technologies, USA) was used to measure total aerosol particle (TAP) and highly fluorescent aerosol particle (HFAP) concentrations, as well as size distributions from 0.5 to 30 µm. The WIBS is a light-induced fluorescence (LIF) sensor that enables the measurement of auto-fluorescence in individual aerosol particles. The instrument operates with an inlet flow rate of $0.3 \, l\min^{-1}$, directing air through an aerodynamic lens. A 635 nm laser measures particle size and shape and serves as a trigger for two UV-Xenon lamps. These lamps excite particles

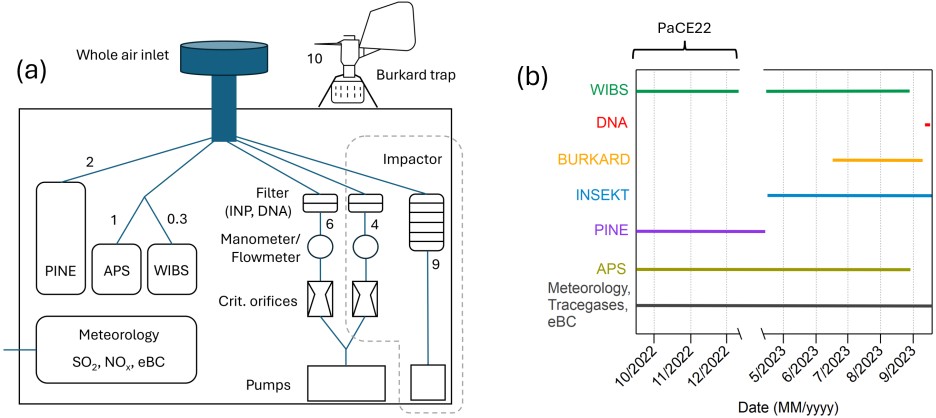

**Figure 1.** (a) Setup of used instruments at Sammaltunturi station. On rooftop: Burkard trap, inside all other instruments. The number beside the instruments indicate the flow rate in $\mathrm{L\,min^{-1}}$. (b) Timeline of when the specific instruments (or filter loading for subsequent analysis) were operated during the campaign.

at wavelengths of 280 nm and 370 nm, with fluorescence subsequently detected by two detectors at wavelength ranges of 310–400 nm and 420–650 nm. This results in three fluorescence channels: FL1 (excitation at 280 nm, emission at 310 - 400 nm), FL2 (excitation at 280 nm, emission at 420-650 nm) and FL3 (excitation at 370 nm, emission at 420-650 nm). A forced trigger measurement is used to identify the fluorescence background, without any particles present in the scattering volume. The forced trigger measurement was done every 6 h for 1 minute to account for a possible shift in the background. Here, an aerosol particle is labeled as highly fluorescent (HFAP) if it exceeds the threshold $TH$, calculated by

$$TH = \overline{FT} + 9\sigma \tag{1}$$

in any channel, where $\overline{FT}$ is the mean value of the forced trigger intensity and $\sigma$ is the standard deviation of the same. Using Perring nomenclature (Perring et al., 2015), one can define 7 fluorescent particle types, depending on the combination of channels exceeding the threshold: A particles are fluorescent in FL1 only, B particles in FL2 only and C particles in FL3 only. AB particles are fluorescent in FL1 and FL2 only, AC particles in FL1 and FL3 only, BC in FL2 and FL3 only and ABC particles are fluorescent in all three channels. The particle types are listed in Tab. 1, along with representative aerosol classes. The approximated shape of particles is calculated via an analysis of the forward scattered light in 4 quarters of the quadrant photo multiplier tube and described by the asymmetry factor $AS$, a dimensionless number between 0 and 100, where a hypothetical perfect sphere should yield the value 0 and and a thin fiber should yield a value close to 100 (Gabey et al., 2010). Data acquisition, processing and quality control is described in detail in Gratzl et al. (2025) for data collected during the PaCE22 campaign, and was handled in the same way for the second part of the campaign.

**Table 1.** Description of WIBS particle types and their corresponding fluorescence channels, along with representative particle classes iden-tifies using a WIBS-5, WIBS-NEO or WIBS-5/NEO. Earlier WIBS models operated with different detector gain settings. A comprehensive overview covering all WIBS models is provided in Gratzl et al. (2025). (a): Katsivela et al. (2025), $9\sigma$ threshold; (b): Gratzl et al. (2025), 3 and $9\sigma$ threshold (c): Beck et al. (2024); $9\sigma$ threshold; (d) Gao et al. (2024), $9\sigma$ threshold; (e): Stone et al. (2021), threshold not reported; (f): Mampage et al. (2022); $3\times\overline{FT}$ threshold; (g): Hughes et al. (2020), $3\times\overline{FT}$ threshold; (h): Clancy et al. (2025), 6 and $9\sigma$ threshold; (i): Sarangi et al. (2022), threshold after Perring et al. (2015). SPP refers to sub-pollen particles.

| Type | Active channels | Representative aerosol classes |
|------|-----------------|--------------------------------|
| A | FL1 only | bacteria[a], microplastics[b] |
| B | FL2 only | eBC[c,d], pollen fragments/SPP[e,f,g] |
| C | FL3 only | fungal spores[h] |
| AB | FL1 and FL2 only | pollen fragments/SPP[f], microplastics[b] |
| AC | FL1 and FL3 only | fungal spores[i] |
| BC | FL2 and FL3 only | eBC[c,d], pollen[h], fungal spores[f,h], pollen fragments/SPP[e,f] |
| ABC | FL1, FL2 and FL3 | fungal spores[f,h,i], pollen[b], pollen fragments/SPP[f], microplastics[b] |

## 2.4 Ice nucleating particle measurements

### 2.4.1 INSEKT

The Ice Nucleation Spectrometer of the Karlsruhe Institute of Technology (INSEKT) is an offline-based instrument to investi-gate the freezing ability of aerosol particles down to -25 °C. 47 mm polycarbonate Nuclepore filters (111137, Whatman®) with 0.2 $\mu$m pore size were pre-cleaned with 10 % $H_2O_2$ and afterwards rinsed with deionized water that was passed through a 0.1 µm syringe filter (6784-2501, Whatman®). The filters were installed in a custom-made stainless steel filter holder. Loaded fil-ters were removed from the filter housing, placed into sterile petri dishes and stored in a freezer. The filter handling is described in detail in Böhmländer et al. (2024). The standard flow rate was held constant at 6.0 L min$^{-1}$ using a critical orifice and a pump (SH-110, Agilent Technologies, Inc). The critical orifice was fabricated by drilling a small hole through a closed ISO-FK sealing. The flow rate was verified using a flowmeter (Defender 530+, Mesa Labs, Inc.). Sampling times were typically one week but ranged from 6 to 17 h for certain filters at the beginning and end of the second part of the campaign. Consequently, the sampled air volume was approximately 60 m$^3$ for most filters. Filters were transported to Germany in a cooled styrofoam box and upon arrival stored in a freezer at -20 °C until analysis. Aerosol suspensions are created by inserting loaded filters into a polycarbonate tube filled with Nanopure™ water (5 or 8 mL) and subsequent rotating (Schneider et al., 2021). The aerosol suspension and dilutions are split into 50 µL aliquots and filled into PCR trays. Each INSEKT run consists of two 96-well plates (192 wells total), with 32 wells reserved for Nanopure™ water as backgrounds. The remaining wells are divided between one or two samples, depending on the number of dilutions used. The PCR trays are cooled at a rate of 0.33 K min$^{-1}$

and the greyscale of each aliquot is observed with a camera. A sudden drop in the greyscale value signals the occurrence of a freezing event, leading to a fraction of frozen aliquots as a function of temperature.

The INP concentration in suspension $C_{INP,sus}$ is calculated from the frozen fraction according to Vali (1971):

$$C_{INP,sus} = -\frac{d}{V_{well}} \cdot ln(f_l),$$

(2)

where $d$ is the dilution scale, $V_{well}$ is the volume of one PCR well and $f_l$ is the fraction of of liquid droplets. To calculate the INP concentration per sampled air volume $C_{INP,air}$, we use

$$C_{INP,air} = \frac{V_{sus}}{V_{air}} \cdot C_{INP,sus},$$

(3)

where $V_{sus}$ is the volume of the whole suspension and $V_{air}$ is the volume of the sampled air. The uncertainties for the INP concentration follow the calculation by Agresti and Coull (1998) for a 95 % confidence interval. The uncertainty for the temperature is one standard deviation of the mean. A Nanopure water background is always present in each experiment, limiting the minimal temperature to -25°C. INSEKT is based on the Colorado State University - Ice Spectrometer (Hill et al., 2016) and additional technical details are given in Schneider et al. (2021).

The aerosol suspensions were - in addition to a standard analysis - subjected to a heat treatment to disable the ice nucleating ability of heat-labile INPs at 95 °C for 20 minutes. These heat-labile INPs are typically of proteinaceous origin, allowing a better understanding of the INP population present during filter sampling Schneider et al. (2021).

### 2.4.2 PINE

The Portable Ice Nucleation Experiment (PINE) is an expansion-type cloud chamber with a volume of 10 L (Möhler et al., 2021). Inside an expansion-type cloud chamber, the pressure is decreased rapidly, leading to a near-adiabatic decrease in temperature due to expansion cooling, while the relative humidity (R.H.) increases. If aerosol particles are present, they are able to act as CCN and/or as INP, leading to the formation of cloud droplets and/or ice crystals. The chamber can be operated between -60 and -10 °C to measure in the mixed-phase regime (> -35 °C) as well as the cirrus regime (< -35 °C). During the PaCE22 campaign, the PINE was either operated in temperature ramp mode, or at constant temperature ranging between -22 °C and -31 °C. The cloud droplets and ice crystals are detected optically with an optical particle counter (OPC). During analysis they are divided by their optical size, and the number of ice crystals is counted, representing the number of INPs. An expansion is performed every six minutes, leading to a high temporal resolution of the INP concentration. The PINE is an autonomous instrument that can be controlled remotely via a custom-made LabVIEW program. This dataset is described in more detail in Böhmländer et al. (2025).

### 2.5 Burkard Spore trap

Pollen and fungal spores were monitored with a Burkard 7-day volumetric sampler (short Burkard trap, Burkard Manufacturing Co. Ltd., UK; (Hirst, 1952)). The sampler was located on an open corridor on the rooftop of Sammaltunturi station approximately 3 m above ground. The sampling and analyzing methods followed the standard methodology adopted by the Finnish

pollen information network following the principles of the European Aeroallergen Network (https://ean.polleninfo.eu/info/en/). The air suction volume of the Burkard sampler was, on average, $10 \, \mathrm{L \, min^{-1}}$. Airborne particles were trapped on a Melinex tape coated with an adhesive (Vaseline and paraffin wax mixture). The tape was attached to a clockwork device, which rotated at the rate of $2 \, \mathrm{mm \, h^{-1}}$. After sampling, the tape was cut into pieces, each corresponding to a single day of sampling under normal operation. Due to occasional interruptions, however, data is not available for every day. These pieces were mounted on slides with Gelvatol under a cover glass. Pollen grains and some plant and fungal spores were identified based on morphological features and counted from the samples using optical microscopy. Pollen grains were identified to Family or Genus level. Spores were counted, if the identification to the taxonomic group, where the recognition could be done reliably, was possible. These are hereinafter referred to as identifiable fungal spores and identifiable plant spores. The sum of both identifiable fungal spores and identifiable plant spores are referred to as identifiable spores. In addition, from three separate days, total number of all fungal spores were counted (even if they could not be assigned to any higher taxonomic level). The spores were divided into two groups: a) particles that could be reliably identified to be fungal spores and b) a group where the identification was slightly uncertain, but particles were believed to be fungal spores. Both together are referred to as total fungal spores. The concentrations of the pollen grains, plant spores and fungal spores of each taxa were determined by random sampling of microscopic fields, with a time accuracy of $2 \, \mathrm{h}$ (Mäkinen, 1981). In the end, the pollen and spore counts were converted to daily averages per $\mathrm{m^3}$ of air.

## 2.6  eDNA Sequencing

For environmental DNA (eDNA) sequencing of airborne fungi, 47 mm polycarbonat filters (Whatman®) with 0.2 µm pore size were installed in a metal filter housing and changed twice a day for 4 days, resulting in 8 filters. Loaded filters were removed and stored in a freezer. The filter holder was cleaned carefully with Isopropanol before a new filter was installed. The flowrate was recorded at all times and held constant at $4 \, \mathrm{L \, min^{-1}}$ using a critical orifice and a pump. Filters were transported to Austria in a cooling box and instantly upon arrival put in a freezer at -20 °C, until brought to the AIT Austrian Institute of Technology GmbH for eDNA analysis.

For DNA extraction filters were dissolved in 750 µL Phenol/Chloroform/Isoamylalcohol solution. Subsequently, 750 µL buffer AP1 (proprietary Qiagen sample lysis buffer) from Plant Mini-Kit (Qiagen) was added to the organic solvents. The mixture was vigorously vortexed and centrifuged at $13000 \, \mathrm{rpm}$ for $10 \, \mathrm{min}$. The aqueous phase was removed and mixed with 400 µL P3 (proprietary Qiagen precipitation buffer). All further steps followed the manufacturer's protocol. At the end, DNA was eluted with 50 µL AE (proprietary Qiagen DNA elution buffer). Library preparation and high-throughput sequencing was done at LGC Genomics GmbH (Berlin, Germany). The fungal ITS2-region (subregion in ITS between genes for 5.8S and 18S ribosomal RNA; ITS: internal transcribed spacer, region between genes for ribosomal RNA) was amplified with primer pair ITS3Mix/ITS4Mix (adapted from Tedersoo et al. (2014) as outlined in Gorfer et al. (2021)) containing Illumina TrueSeq adapters. Reaction mixtures contained 15 p mol of each forward and reverse primer in 20 µL volume of $1 \times$ MyTaq buffer containing 1.5 units MyTaq DNA polymerase (Bioline GmbH) and 2 µL of BioStabII PCR Enhancer (Sigma-Aldrich). The first amplification of the two-step PCR approach ran for 30 cycles: 1 min 96 °C pre-denaturation followed by 15 s at 96 °C

denaturation, 30 s at 58 °C annealing and 90 s at 70 °C extension and a final hold at 8 °C. The second amplification was as the previous amplification but with standard i7- and i5- sequencing adapters and with modified annealing temperature, i.e. 3 cycles at 50 °C followed by 7 cycles at 58 °C. DNA concentration of amplicons was assessed by agarose gel electrophoresis. About 20 ng of indexed amplicon DNA of each sample was subsequently pooled. The pooled libraries were purified with one volume of Agencourt AMPure XP beads (Beckman Coulter) to remove primer dimer and other small mispriming products, followed by an additional purification on MiniElute columns (Qiagen). The size selection was performed by preparative gel electrophoresis on an Low Melting Point Agarose gel. Sequencing was done on an Illumina MiSeq using Illumina MiSeq Reagent Kit v3 (2 × 300 bp (basepairs)). Downstream bioinformatic analyses of raw paired end reads obtained from MiSeq V3 runs were done as previously described (Gorfer et al., 2021). Initial quality filtering was done with Trimmomatic v. 0.39 (Bolger et al., 2014). USEARCH program suite (Edgar, 2010) was used for merging the forward and reverse reads with a minimal overlap of 30 bp with fastq_mergepairs. Sequences < 280 bp were all of non-fungal origin and thus filtered out. FASTX toolkit script fastx_barcode_splitter.pl was used to sort out project-specific fungal sequences; USEARCH scripts were used for chimera detection and filtering underrepresented sequences (< 10). VSEARCH (Rognes et al., 2016) was used for clustering and counting sequences per cluster, using a 97 % sequence similarity, which is a widely used threshold for the ITS region and lies between generally accepted limits for discrimination of species and genera (Vu et al., 2019). OTU (Operational Taxonomic Unit) clustering instead of using 100 % identical amplicon sequence variants (ASVs) was recommended for fungal ITS datasets (Tedersoo et al., 2022). Taxonomic affiliation of OTUs was done with the UTAX script against the UNITE database (Abarenkov et al., 2024). Non-fungal sequences were removed from further analyses. All sequence affiliations were manually evaluated and edited, increasing the phylogenetic accuracy (Hofstetter et al., 2019). Given the set sequence similarity threshold, species names provided for OTUs must be considered sensu lato. Further bioinformatics and statistical analyses were done in R version 4.3.1 (cran.r-project.org) with the following packages: vegan (Oksanen et al., 2018), phyloseq (McMurdie and Holmes, 2013), metagMisc (Mikryukov and Mahé, 2025) and microViz (Barnet, 2024).

Most samples had read numbers for fungi clearly above 3000, but the two samples from the last sampling day had only 815 and 685 reads, respectively. To allow inclusion of these two samples, for further analyses, all samples were rarefied to 685 reads with the function rarefy_even_epth in phyloseq. Retrieval of relative abundances of specific taxa was done by converting the read numbers to relative abundances, agglomerating taxa at the appropriate level (tax_glom in phyloseq) and seleceting desired taxa (subset_taxa in phyloseq) or selecting most abundant taxa (phyloseq_filter_top_axa in metagMisc).

## 2.7 Impactor measurements and fluorescence microscope

An impactor (Sioutas Five-stage Cascade Impactor, SKC, US) was installed down the whole air inlet in parallel to filters for eDNA analysis (see Fig. 1) and was operated at 9 L min$^{-1}$ using a Leland Legacy Personal Sampling Pump (LKC, USA). Sampling time was between 6 and 20 h, with a mean collected volume of 6.5 m$^3$. The flow was checked every day and if necessary calibrated. Aluminum foil (cleaned with aceton and baked at 300 °C for at least 5 h) was installed in four stages (cut-off dimaters: 2.5 μm, 1 μm, 0.5 μm, 0.25 μm aerodynamic diameter (Da)). After each sampling period, the aluminum foils were removed and stored in a freezer. The impactor was cleaned before a new set of foils was installed. The foils were qualitatively

analyzed with a Nikon Eclipse Ci-L microscope (Nikon, Japan) under bright field and fluorescence setting ($\lambda_{ex}$=465-495 nm,
$\lambda_{em}$=515-555 nm), to confirm the presence of fluorescent particles exhibiting morphological properties, typical of fungal spores. Quantitative analysis of fluorescent particles (particle concentration) is discussed in the Supplement Sect. S8.

## 2.8 Meteorological variables and air pollutants

The Sammaltunturi station is equipped with Vaisala Milos 500 automatic weather station. The wind speed was measured with a heated cup anemometer and the wind direction with a heated wind vane. Temperature was measured with Pt100 sensors and R.H. with a Vaisala sensor (HUMICAP). Visibility was measured by a Vaisala FD12P present weather sensor. Data is saved as one minute means (Hatakka et al., 2003).

The equivalent Black Carbon (eBC) concentration in Sammaltunturi is measured optically with an AE-33 Aethalometer (Magee Scientific) instrument connected to the whole air inlet. The instrument measures how light is attenuated as it passes through a filter loaded with particles, which converts to aerosol absorption. The aerosol absorption is used to estimate the eBC mass, assuming a default, constant mass absorption cross-section (MAC) of $7.77 \, \mathrm{m^2 \, g^{-1}}$ at 880 nm and a filter-tape correction of 1.39 (Drinovec et al., 2015). AE33 features a dual-spot configuration, analyzing two filter spots with differing airflow rates at the same time. Additionally, a reference spot, unaffected by the sample flow, is used to monitor any fluctuations in the light source's intensity. To ensure measurement reliability, the instrument performs regular zero-flow and optical calibrations. Once the attenuation value reaches 120, it automatically advances to a filter spot.

The gas compositions in Sammaltunturi are measured through a common sampling inlet line with a flow of about $90 \, \mathrm{m^3 \, h^{-1}}$ and a residence time of two seconds (Hatakka et al., 2003). A chemiluminescence analyzer with molybdenum converter, Thermo Environmental Instruments 42 CTL (TEI42CTL), is used to measure $NO_2$ concentrations. At Sammaltunturi station, the $NO_2$ concentration measured by TEI42CTL is a sum of all oxidized nitrogen species. However, $NO_x$ concentration levels at Pallas are low, especially during summer. Thus, the results are most of the time below the instrument detection limit (50 ppt). Analysis of $SO_2$ is based on measuring UV fluorescence. The instrument used is Thermo Environmental Instruments 43S. Details on air pollutant monitoring can be found in Hatakka et al. (2003).

## 2.9 Air mass back trajectories

Air mass back trajectories were carried out using the NOAA HYSPLIT Trajectory Model v5.3.0 (Stein et al., 2015; Rolph et al., 2017), using meteorological data from the Global Data Assimilation System (GDAS) archive (1 degree resolution). 72 h back trajectories were calculated for air masses arriving either 100 m or 500 m above ground level at Sammaltunturi station.

## 3 Results and Discussion

### 3.1 The contributions of different HFAPs change seasonally

Figure 2 provides an overview of the monthly fractions of HFAPs within TAPs, ambient temperature (measured at Sammaltunturi), and and the relative contributions of different fluorescent particle types to HFAPs. The fractions of HFAPs exhibit a

clear seasonal trend. At the start of the PaCe22 campaign in September 2022, the median fraction is $5.6 \times 10^{-2}$ and steadily decreases until the end of the PaCE22 campaign in December 2022, when the minimum monthly median value of $6.5 \times 10^{-3}$ is reached. From April to August 2023, the HFAP fraction increases continuously, peaking in August at $7.2 \times 10^{-2}$. The logarithm of the fraction roughly follows ambient temperature trends (depicted in red), with minima and maxima occuring in December 2022 and August 2023, respectively. Figure 2 (b), illustrates the montly contributions of different fluorescent particle types

to HFAPs. The fractions of AB and ABC particles follow the seasonal trend of temperature and total HFAP fraction. During warmer months, AB and ABC particles dominate, accounting for 60 % of HFAPs in September, 67 % in July and 75 % in August. In contrast, their contribution drops to 12 % in December. Conversely, B particles (and to a lesser extent BC particles) show an inverse pattern, dominating during colder months. The contribution of A particles does not exhibit a clear seasonal trend. In the following, we will focus on AB and ABC particles, since their high contribution during warmer months suggests

a possible biological source. The contributions of C and AC particles are negligibly small and will not be discussed further. (Details on A, B, and BC particles are provided in the Supplement in Sect. S2).

### 3.2 Local snow cover influences the concentration of biological HFAPs

Figure 3 shows the daily mean concentrations of TAPs, HFAPs, AB, and ABC particles. Snow depth and ambient temperature, measured at the Kenttärova research station (approximately 5.5 km east of Sammaltunturi), are also depicted in Figure

3 (a). The temperature is used to define meteorological seasons (see Supplement Sect. S2 for seasonal definitions, particle concentrations, and size distributions of all fluorescent particle types).

TAPs show no clear seasonal trend (Fig. 3 (b)). The lowest median TAP concentration occurs in autumn ($0.28$ cm$^{-3}$), while the highest is in summer ($0.75$ cm$^{-3}$). In contrast, HFAP concentrations exhibit a strong seasonal cycle, peaking in summer with a median of $3.0 \times 10^{-2}$ cm$^{-3}$, approximately 10 times higher than in winter. Winter and spring peaks of HFAPs

are primarily due to B and BC particles, with A particles playing a minor role (details in Supplement Sect. S2). AB and ABC particle concentrations follow a similar pattern, suggesting a common biological origin. These particle types show the strongest seasonal cycle, with minimum median concentrations in winter ($2.0 \times 10^{-4}$ cm$^{-3}$) and maxima in summer ($5.8 \times 10^{-3}$ cm$^{-3}$ and $9.5 \times 10^{-3}$ cm$^{-3}$ for AB and ABC, respectively).

Snow cover significantly affects AB and ABC particle concentrations. Once the ground is covered in snow (from October 23

to May 22), concentrations decrease rapidly and remain low. For instance, the median ABC concentration in autumn ($4.4 \times 10^{-3}$ cm$^{-3}$) drops 22-fold in winter. This strongly suggests, that snow cover suppresses the emission of AB and ABC particles from the local biosphere. Once the snow begins to melt, concentrations gradually increase. Peaks in AB and ABC concentrations during the melting period correspond to air masses passing over snow-free land north of the Gulf of Bothnia, such as Ylitornio

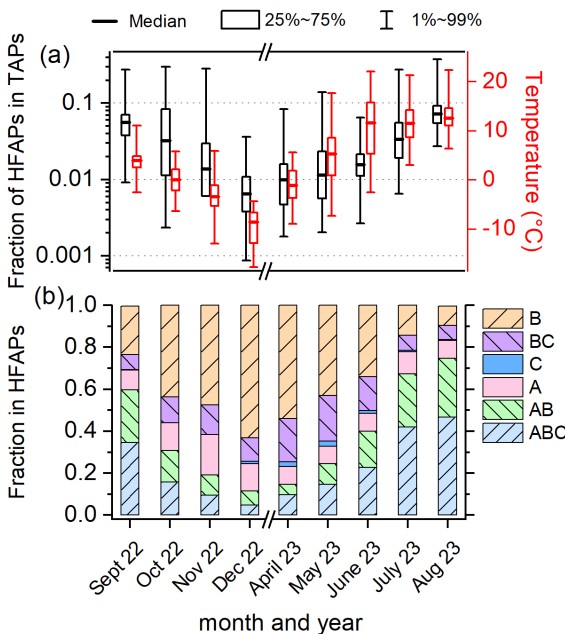

**Figure 2.** Overview of fraction of highy fluorescent aerosol particles (HFAPs) and ambient temperature. (a) Monthly box diagrams (median, 25th to 75th percentile and first to 99th percentile) of the fraction of total HFAPs in total aerosol particles (TAPs) and temperature. Boxplots are based on hourly mean values. (b) Monthly contributions of HFAP types to total HFAP concentration (fraction of median values). AB and ABC particles dominate in the warmer months, while B and BC particles dominate in the colder months. The fraction of AC particles is not shown, as they contributed only 0.01 % or less to HFAP concentrations.

Meltosjärvi (66.5333°E, 24.65° E). This suggests that regional biosphere emissions south of Sammaltunturi contribute to increased concentrations (details in Supplement Sect. S3.3). A notable example occurred on November 5, 2022 (labeled (1) in Fig. 3). On this day—the third in a three-day melting period—snow depth (measured at 8 am local time) decreased from $9.0 \pm 2.0$ cm to $3.3 \pm 2.0$ cm. Air masses on this day arrived from the south and spent most of the time close to the ground, following similar trajectories to the days before and after. However, in areas slightly south of Pallas, such as Kolari Kattilamaa (70 km south), snow had completely melted. This lead to an sudden 10-fold increase in ABC concentrations compared to previous two days, indicating that local emissions of AB and ABC particles resumed immediately after snow melt (details in Supplement Sect. S3.1). Days of maximum AB and ABC concentrations always correspond to local air masses near the ground (e.g., August 2, which recorded the highest daily mean value, labeled (2) in Fig. 3, see Supplement Sect. S3.4 for details). This further supports the hypothesis that AB and ABC particles originate mostly from the local biosphere. Potential local sources, even under snow-covered conditions, include cellulolytic fungi such as certain Polyporales species which grow on trees and can produce fruitbodies that persist year-round. They have been shown to sporulate under cold and harsh conditions (Gonthier et al., 2005). Aerosolized soredia from tree-dwelling lichens (Armstrong, 1991) and spores from mosses (Ščevková et al.,

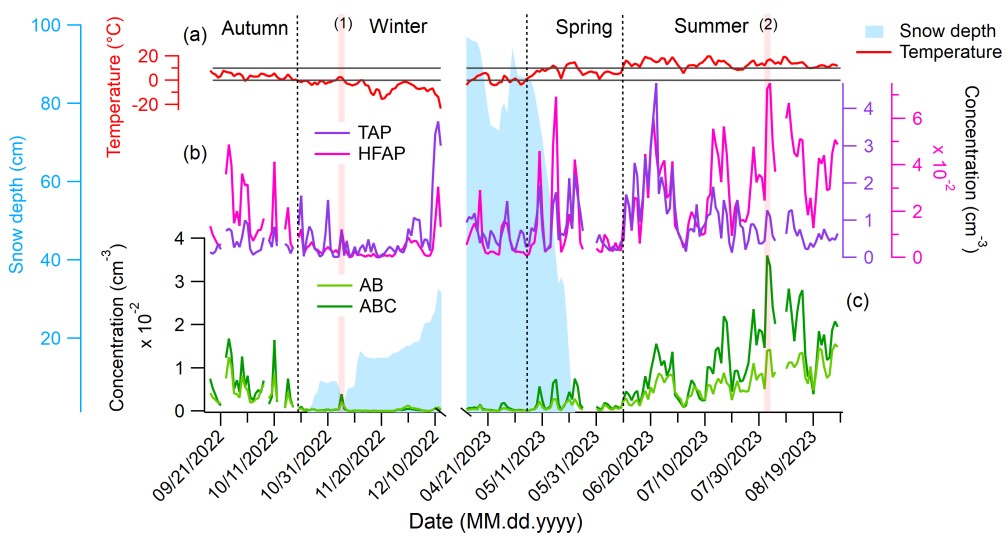

**Figure 3.** Daily mean concentrations of TAPs and HFAPs, as well as AB and ABC particle types, mean snow depth and ambient temperature inside a near spruce forest. (a) Temperature measured at Kenttärova station is used to define the meteorological seasons. Grey lines are at 0 °C and 10 °C. (b) TAPs and HFAPs, (c) AB and ABC particles. Vertical transparent red lines are labeled with a number and mark the two days discussed in the text. Snow cover reduces the concentration of AB and ABC particles drastically.

2024) growing on tree trunks or rocks may also contribute. The fact that these organisms grow on trees, above the snow pack, may facilitate PBAP emissions even when the ground is snow-covered.

Similar relationships between fluorescent particles and snow cover have been reported in the boreal forest. Schumacher et al. (2013) observed this phenomenon in Hyytiälä, southern Finland, using a UV-APS (TSI, USA) instrument with a fluorescence channel similar to FL3 in WIBS. At the same site, Schneider et al. (2021) found that AB and ABC particles exhibited the strongest seasonal trends, strongly influenced by snow cover. Petersson Sjögren et al. (2023) reported a comparable seasonal trend in a managed coniferous forest at the Hyltemossa research site in southern Sweden using a Biotrak sampler (TSI, USA). This instrument, which operates with similar excitation and emission wavelengths ($\lambda_{ex} = 405$ nm, $\lambda_{em} = 405 - 600$ nm, D > 1 μm) as the FL3 channel in WIBS, recorded concentrations similar to those measured in this study. Since C and AC concentrations are negligible small, FL3 is primarily composed of BC and ABC particles. In autumn, the concentrations in Hyltemossa and Pallas (FL3) are $5 \times 10^{-3}$ and $6 \times 10^{-3}$ cm$^{-3}$, respectively. In winter, they are $3 \times 10^{-3}$ and $7 \times 10^{-4}$ cm$^{-3}$, in spring, they are $3 \times 10^{-3}$ and $2 \times 10^{-3}$ cm$^{-3}$ and in summer, they are $1.3 \times 10^{-2}$ and $1.4 \times 10^{-2}$ cm$^{-3}$.

### 3.3 Diurnal patterns reveal the coupling to the local surface

Summer-time diurnal median values were calculated only for periods when the station was outside of clouds (defined as visibility > 1000 m), as in-cloud measurements result in a considerable reduction of TAPs and HFAPs (with the former declining

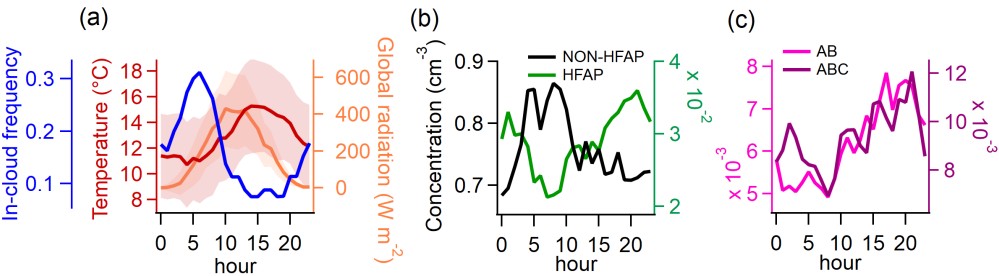

**Figure 4.** Diurnal patterns of meteorological parameters and particle concentrations in summer for times in which the station was not exposed to clouds. (a) Temperature, global radiation and frequency (or probability) of the station being inside a cloud (used as a proxy for probability of the station being in the free troposphere). (b) Concentration of NON-HFAPs and HFAPS. NON-HFAPs can also contain FAPs that did not classify as HFAPs. (c) Concentration of AB and ABC particles. Similarities between in-cloud frequency and NON-HFAP concentration suggest a higher contribution of far-range transport to NON-HFAP concentration. The opposite behavior of HFAP concentrations suggests locally sourced bioaerosols are the major contributors to HFAP concentrations.

much more sharply, see Supplement Sect. S4). The frequency of the station being inside clouds follows a distinct diurnal cycle (Fig. 4, (a), blue line) peaking at 5 am and reaching a minimum between 2 pm and 7 pm. This diurnal pattern of in-cloud frequency provides insights into whether the station is inside or outside the mixing layer at different times of the day. When
cloud frequency is high at Sammaltunturi station (565 m), it is more likely that the station is inside or above the boundary layer and therefore decoupled from the local biosphere. This aligns with findings by Aalto et al. (2002), who reported temperature inversions between Pallasjärvi station (303 m) and Sammaltunturi from 9 pm to 6 am, whereas daytime typically lacked inversions. Data from the Sodankylä observatory, 125 km southeast of Pallas, show that the mixing layer in summer afternoons usually extends to 1000-1500 m (Aalto et al., 2002).

The diurnal cycle of NON-HFAPs (TAPS - HFAPs) (Fig. 4 (b)) roughly follows in-cloud frequency trend, suggesting that the highest concentrations occur when the station is in the free troposphere. In contrast, HFAP concentrations (Fig. 4 (b)) increase steadily from 7 am, peaking at 9 pm, indicating that HFAPs are most abundant when the station is within the mixing layer. A similar diurnal pattern was observed by Petersson Sjögren et al. (2023) in a coniferous forest in the south of Sweden. AB and ABC particle concentrations (Fig. 4 (c)) show a delayed increase with temperature, peaking between 3 pm and 9 pm,
following the general trend of HFAPs. This further supports the assumption that PBAPs originate from the local surface, with long-range transport playing a minor role in PBAP distribution. Similarly, Gao et al. (2024) found significantly higher HFAP concentrations within the planetary boundary layer than in the free troposphere at Mount Helmos in the eastern Mediterranean.

### 3.4 HFAPs are influenced by meteorology and air pollution

To determine whether HFAP concentrations are primarily controlled by natural meteorological variables or anthropogenic
influences, Pearson correlation analysis was conducted on hourly mean WIBS data, meteorological parameters (temperature,

R.H.), eBC, and $NO_x$. Results are shown in Fig. 5 (a), with only Pearson r-values (from now on r) corresponding to p-values < 0.05 presented.

TAP concentrations show no strong correlation with temperature, R.H. or $NO_x$, but exhibit a correlation with eBC mass concentration (r = 0.6). Similarly, HFAP concentrations correlate with eBC concentrations (r = 0.61) but also show a weak positive correlation with R.H. and a moderate correlation with $NO_x$ (r = 0.42) for out-of-cloud periods. HFAP concentrations also correlate with temperature (r = 0.61).

AB and ABC concentrations show correlations of 0.60 and 0.59, respectively, with temperature. This correlation strengthens when considering the common logarithm of ABC particle concentration, yielding an r-value of 0.82 (p < 0.0001) (Fig. 5 (b)). A similar logarithmic relationship is observed for AB particles, with an r-value of 0.78 (p < 0.0001). This relationship aligns with previous findings that biological activity in the atmosphere increases with temperature over seasonal time scales (e.g. Pereira Freitas et al. (2023); Petersson Sjögren et al. (2023); Schumacher et al. (2013)). A weak positive correlation between R.H. and ABC (r = 0.35) and AB concentrations (r = 0.27) is observed for out-of-cloud periods. Several studies have reported increases in bioaerosol concentrations during and after rain events (Rathnayake et al., 2017; Yue et al., 2016; Gosselin et al., 2016; Heo et al., 2014; Schumacher et al., 2013; Huffman et al., 2013; Prenni et al., 2013; Allitt, 2000; Hirst and Stedman, 1963; Gregory and Hirst, 1957). In contrast, no such relationship was observed in our study between rainfall and increased concentrations of HFPAs. A likely explanation is that, during most rain events recorded at our station, the site was simultaneously within a cloud. These in-cloud conditions led to a marked decrease in the concentrations of both TAPs and HFAPs due to cloud and rain scavenging (Flossmann and Wobrock, 2010; Sellegri et al., 2003). However, HFAPs, particularly the AB and ABC subtypes, showed a much smaller decline compared to TAPs, resulting in a substantial relative increase in the AB and ABC fractions during in-cloud events. Further details on the effect of rain and cloud immersion on the concentrations and fractions of HFAPs and TAPs are discussed in the Supplement Sect. S4.

Only weak correlation were found between AB and ABC concentrations with eBC and $NO_x$ concentrations, suggesting that air pollution has minimal influence on the concentration of these particle types compared to temperature. However, B and BC concentrations show the strongest correlation with eBC mass concentration (0.71 and 0.65, respectively). The correlation increases when using daily means (r = 0.82). Additionally, B and BC concentrations have stronger correlations with $NO_x$ than AB and ABC particles, indicating a greater influence from anthropogenic pollution. This strong correlation between either B or BC concentrations with eBC has been reported a few times in literature (Beck et al., 2024; Markey et al., 2024; Gao et al., 2024; Yue et al., 2022). Therefore, B and BC particles are considered to be mostly of anthropologic origin. See Supplement Sect. S2 for more details of B and BC particles.

To further analyze the relationship between ABC particles and meteorological parameters, data were divided by season, considering only out-of-cloud periods. R.H., temperature and wind speed were binned to channels with widths of 5 %, 2 °C and $2\,\mathrm{m\,s^{-1}}$, respectively. Median values of hourly mean concentrations within each bin were calculated, if more than 10 values exist for that bin.

In spring and winter, R.H. has little effect on ABC particle concentration, whereas in summer, a positive relationship appears, with maximum median concentrations occurring at 80-100 % R.H. (Fig. 6 (a)). A similar but weaker trend is observed in

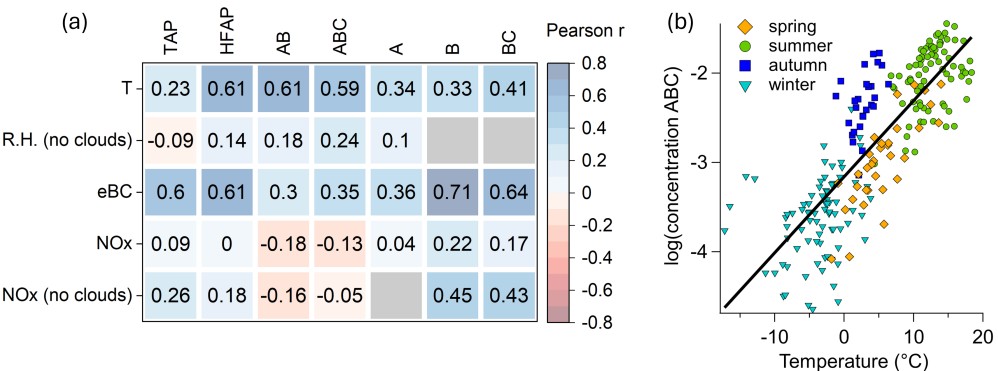

**Figure 5.** Relation of concentration of particles types to meteorological parameters and air pollution. (a) Pearson correlation coefficient matrix of 1 h mean values. Grey (empty) relations do not reach the 95 % confidence level. (b) scatter plot of the common logarithm of daily mean ABC particles against ambient temperature (r = 0.82). The transition between snow free and snow-covered periods (e.g., from autumn to winter) is less apparent than in Fig. 3 (b), as the temporal context is lost and ABC particles remain present to varying extents throughout the winter.

autumn. In spring, summer and autumn, ABC concentrations increase with temperature up to 15 °C but decline at higher temperatures in summer (Fig. 6 (b)). Wind speed has an effect in summer, with peak median concentrations on calm days (< 4 m s$^{-1}$), while in winter, there is a slight positive correlation between ABC concentration and increasing wind speed (Fig. 6 (c)). The maximum concentrations at high R.H. and the increase with temperature up to approximately 15 °C, with a decline at higher temperatures, may reflect a fungal spore release mechanism (e.g., surface tension catapult; Pringle et al. (2005)), which favors high humidity and moderate temperatures. The high fungal activity in autumn could also be responsible for the higher ABC concentrations observed already at lower temperatures compared to summer. In the same sense, in spring, fungal activity is lower, resulting in almost no R.H. dependence. Notably, the highest concentrations in spring are observed when the surrounding ground is still snow-covered (see Fig. 3), and when air masses originate from the south (see Supplement Sect. S3.3), where snow melt has already occurred. This suggests that PBAPs may be transported to Pallas with these warmer air masses (and often lower R.H.). In winter, most local sources are buried under snow and no trend with R.H. and temperature is observed at the overall low concentrations of ABC particles. However, the slight positive trend between ABC concentrations and wind speed suggest that the few potential local sources such as cellulolytic fungi, lichens, or mosses (see Sect. 3.2), may be more efficiently aerosolized under stronger winds (Ščevková et al., 2024; Armstrong, 1991; Kallio, 1970), transported to the site, or result from biological residues in blowing snow (Boetius et al., 2015).

In summary, ABC concentrations peak under high R.H., moderate temperatures (10–15 °C), and low wind speeds in summer. In autumn, both temperature and RH positively influence concentrations, although wind speed plays a lesser role. In spring, temperature (up to 15 °C) is the dominant factor, while in winter, wind speed is the only meteorological variable with a clear

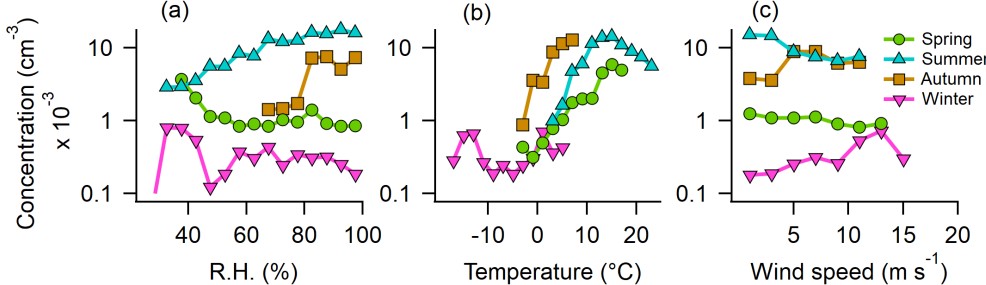

**Figure 6.** The median concentration of ABC particles dependent on meteorological parameters for each season. Depicted are median concentration values for 1 h mean values. (a) R.H., (b) ambient Temperature, (c) wind speed.

impact. Similar seasonal patterns are observed for AB particles (see Supplement Sect. S9), whereas B and BC particles show
no clear relationship with R.H. but tend to increase with temperature in both summer and spring.

## 3.5 ABC particles are dominated by fungal spores

From June 14, 2023, a Burkard trap was installed on the rooftop of Sammaltunturi station to collect airborne pollen grains
and fungal spores for microscopic identification. To further differentiate WIBS data and to compare it with Burkard trap
measurements, we analyzed the absolute fluorescence intensity in the FL1 channel of each detected ABC particle during the
second part of the campaign. We found a multi-modal intensity distribution and identified three distinct fluorescence intensity
regions (compare with Clancy et al. (2025), who observed a bimodal distribution). We defined a new FL1 threshold at $1.0 \times 10^{9}$
intensity units, based on a local minimum of the trimodal intensity distribution (see Supplement Sect. S5, Fig. S10). ABC
particles exceeding this new threshold were labeled ABC_3, while all other ABC particles were labeled ABC*. It should
be stressed, that the fluorescence threshold used to discriminate ABC* and ABC_3 in this study is not universal and could
differ strongly from one individual instrument to another. More details on the trimodal intensity distribution is provided in the
Supplement in Sect. S5.

The majority of pollen grains identified in the Burkard trap were pine (*Pinus*, 67 %) and from the cypress familiy (*Cupressaceae*, 12 %) of which most were detected before July. Figure 7 (a, top), shows the total pollen, pine pollen and ABC_3
concentrations, revealing a highly similar timely evolution. A scatter plot of ABC_3 versus pine pollen concentration (Fig. 7
(b, left) shows a strong correlation (r = 0.92, p < 0.0001), although it is acknowledged that the number of data points above
400 m$^{-3}$ is limited. Similarly, ABC_3 and total pollen concentration have an r-value of 0.90. The absolute concentrations of
both ABC_3 and pine pollen are comparable ($\approx 45°$ linear fit).

Figure 7 (a, bottom), shows identifiable fungal spores and identifiable plant spores from the Burkard trap and ABC* particle
concentrations. 90 % of identifiable fungal spores are *Cladosporium*, the remaining 10 % are shared by *Alternaria*, *Pucciniales*
(rust fungi) and smut fungi. Identifiable plant spores were 87 % trilete spores, while the rest consists of *Equisetum*, fern and
*Lycopodium*. A microscopy picture of a Burkard slide with two impacted trilete spores is shown in the Supplement Sect. S6.

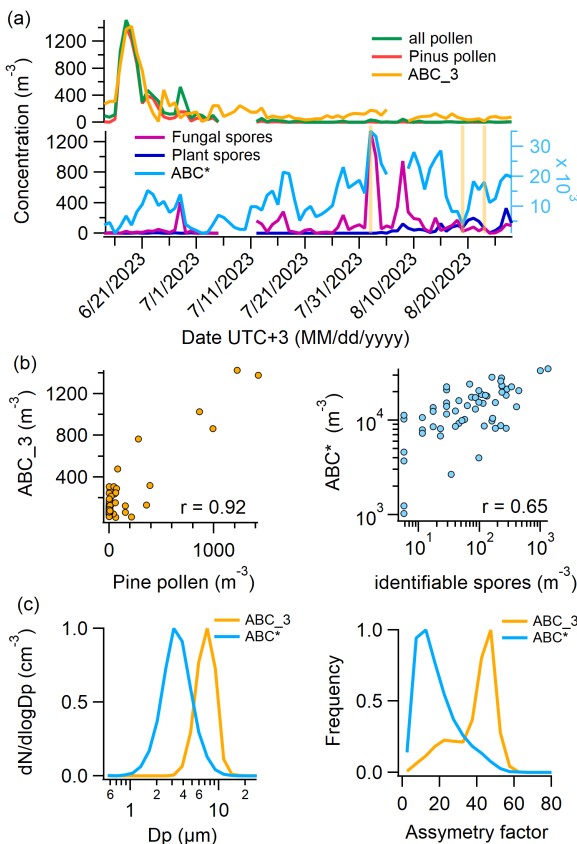

**Figure 7.** Comparison of Burkard- and WIBS data. (a) top: Daily number concentrations of all pollen, pine pollen (which make up 77 % of all pollen), and ABC_3 particles (ABC particles with FL1 fluorescence intensity $> 1 \times 10^9$ in arbitrary units). (a) bottom: Daily number concentrations of identifiable fungal spores, identifiable plant spores and ABC* particles (all ABC particles without ABC_3 particles). (b) left: Scatter plot of ABC_3 vs. pine pollen concentrations. (b) right: Scatter plot of ABC* vs. identifiable spores concentrations (fungal and plant). (c) left: Normalized size distribution of ABC_3 and ABC* particles. (c) right: Normalized asymmetry factor distribution of ABC_3 and ABC* particles.

The correlation between fungal spores and ABC* particles is not as pronounced as with pine pollen and ABC_3 particles, but most of the peaks align. The scatter plot of ABC* and spores (fungal spores and plant spores) (7 (b, right), reveals an r-value of 0.65 (p < 0.0001). Both the correlation between ABC* and pine pollen concentrations, and the correlation between ABC_3
and identifiable spore concentrations are negative, however, only the former is statistically significant (r = –0.23, p < 0.05).

Importantly, r-values only change minimally when using all ABC particles instead of ABC* particles, since the concentration of ABC_3 particles is very low compared to ABC*. The sum of ABC* and AB (as well as ABC and AB) leads to similar r-values as ABC* or ABC alone, namely 0.66. This again is an argument for the same nature (of most) of both ABC and AB particles.

Interestingly, ABC_3 and ABC* particles not only differ significantly in the intensity of their fluorescence, but also in size (Fig. 7 (c), left) and in asymmetry factor (Fig. 7 (c), right). The ABC* size distribution peaks at around 3 μm, whereas ABC_3 particles are exclusively found above 2 μm, with a sharp decline after 8 μm, likely due to losses in the inlet system. The asymmetry factor of ABC* particles is low with a maximum of the distribution at 10. Hence, they are relatively sphere-like. The asymmetry factor of ABC_3 particles, on the other hand, is higher with a maximum of the distribution at 50. The differences in fluorescence, size, shape and relation to pollen grains and fungal spores suggest two distinct particle natures.

Despite the strong correlation between ABC_3 and pollen concentrations, it is unlikely that ABC_3 particles primarily represent pollen grains. Pine pollen, which accounts for 67 % of all counted grains, ranges from 51 to over 100 μm (Halbritter et al., 2021), well above the upper size limit reliably detected by WIBS. Moreover, the WIBS inlet system exhibits substantial losses for particles >10 μm (Gratzl et al., 2025), making the detection of pine pollen highly improbable. Even if a small subset of pine pollen were at or below 10 μm, WIBS would only capture a minor fraction, insufficient to explain the close match in concentrations. Other pollen types also show significant positive correlations ($p < 0.05$) with ABC_3 concentrations, including *Betula* (10–25 μm, r = 0.50), *Cyperaceae* (10–50 μm, r = 0.57), *Picea* (>100 unitμm, r = 0.31), and unidentified pollen types (r = 0.41; size data from https://www.paldat.org/, accessed 19.06.2025). While a small number of these smaller pollen grains, particularly *Betula*, may fall within the WIBS detection range and contribute to ABC_3 counts, their concentrations are too low to account for the total ABC_3 particle abundance observed. In recent years, the presence of sub-pollen particles in the atmosphere has been studied. By the definition of Burkart et al. (2021), these are starch granules which can be expelled from pollen grains during humid conditions and thunderstorms and have been found to fluoresce (partly) in the ABC channel (Hughes et al., 2020; Stone et al., 2021; Mampage et al., 2022). However, they are usually smaller than 2.5 μm in diameter (Burkart et al., 2021; Hughes et al., 2020; Mampage et al., 2022; Suphioglu et al., 1992; Schleh et al., 2010), while ABC_3 particles are larger than 2 μm. A possible explanation could be pollen fragments, as observerd by Hughes et al. (2020), or other plant material that is emitted at the same time as pollen.

Noticeably, the concentration of ABC* particles (and all ABC particles) is much higher than that of identifiable fungal spores with daily averages exceeding spore counts by factors of 20 to 2000. To explain this difference, three days were selected, on which the Burkard data was investigated closer: One of high ABC* concentration (August 2, 2023), one of medium ABC* concentration (August 23, 2023) and one of low ABC concentration (August 19, 2023) (marked as orange vertical lines in the lower subplots of Fig. 7 (a)). For these days, every single particle impacted on the Burkard trap that is definitely or probably a fungal spore (latter contribute between 7 and 13 %), even if it could not be assigned taxonomically, was counted (more than 7000 spores). The resulting daily mean concentration of identifiable fungal spores, total fungal spores (in the Burkard trap), ABC particles as well as AB+ABC for the three days is given in Tab. 2. The concentration of total fungal spores is in the same order of magnitude as the ABC particle concentration, compared to a 1, 2 and 3 orders of magnitude difference between identifiable fungal spores and ABC particles for August 2, 2023, August 19, 2023 and August 23, 2023, respectively. As a result, only a minor fraction of the whole fungal spore population is recorded, when only those fungal spores that can be taxonomically assigned are considered, consistent with Pashley et al. (2012), who found that over 87 % of genera was not identified microscopically). A microscopy picture of the Burkard tape on August 2, 2023 can be seen in the Supplement in

**Table 2.** Concentrations of identifiable fungal spores, total fungal spores (counted on the Burkard tape), ABC particles and the sum of AB and ABC particles on three selected days. On August 2, 19, and 23, ABC concentrations exceed identifiable fungal spore concentrations by approximately one, two, and three orders of magnitude, respectively. Total spores are in the same order of magnitude as ABC and AB+ABC concentrations.

|  | August 2, 2023 | August 19, 2023 | August 23, 2023 |
| --- | --- | --- | --- |
| identifiable fungal spores ($m^{-3}$) | 1339 | 23 | 69 |
| total fungal spores ($m^{-3}$) | 14466 | 6132 | 22119 |
| ABC particles ($m^{-3}$) | 35359 | 4046 | 18086 |
| AB+ABC particles ($m^{-3}$) | 48538 | 8995 | 28670 |

Sect. S6. The sum of AB and ABC particles still is considerable higher than that of total fungal spores, especially on August 2. The Burkard trap likely underestimates smaller spores due to strongly increasing difficulties to correctly identify particles as fungal spores, especially colorless (hyaline) ones, with decreasing size. Furthermore, despite being used in aerobiological studies since more than 70 years, to the best knowledge of the authors, a characterization of the detection efficiency of a Hirst type spore trap has never been done for particles < 10 μm. On the other hand, the WIBS (at least in the present study) fails to correctly count the bigger fungal spores due to inlet losses. Surprisingly, both methods give comparable concentration, at least on days with medium and low ABC concentrations. In a similar boreal forest environment Helin et al. (2017) found that airborne fungal DNA was predominantly found in the particle size range < 10 μm, and Abrego et al. (2024) found that the community-weighted mean spore size of continental polar fungal spores is below 5 μm. This highlights the importance of fungal spore measurements in the smaller size range.

Most field studies also found that the highest correlation with fungal spores was achieved with ABC particles (Sarangi et al., 2022; Fernández-Rodríguez et al., 2018). Similar results were found when comparing WIBS data with fungal spore tracers (Yue et al., 2019, 2022; Gosselin et al., 2016). In chamber studies, however, fungal spores have been characterized almost exclusively as A particles (Hernandez et al., 2016; Savage et al., 2017). In our study, A particles showed no significant correlation to identifiable fungal spore counts (differing to Markey et al. (2022)). This discrepancy could have two main reasons. (a): Atmospheric aging (e.g., UV radiation, water uptake, other atmospheric components (Pan et al., 2021)) may alter fluorescence properties. For instance, fluorescence of *Aspergillus niger*, a common mold fungus, was strongly enhanced after UV-light exposure (Raimondi et al., 2009). This could lead to stronger fluorescence in the FL3 channel for air-exposed fungal spores compared to laboratory grown cultures and therefore to an enhanced ABC signal. (b): There could be significant instrument variability of obtained fluorescence signals (Hernandez et al., 2016), which might even be enhanced when comparing different model versions of the WIBS.

Some studies report poor correlation between WIBS data and *Cladosporium* counts (Healy et al., 2014; O'Connor et al., 2015), possibly due to the fact that *Cladosporium* spores can get released in larger clusters which either get counted as one single particle by the WIBS or is lost in the inlet (Markey et al., 2022). However, in this study, *Cladosporium* spores dominate

identifiable fungal spores and drive peak ABC* concentrations (Fig. 7 (a, bottom)). Subtracting *Cladosporium* counts reduces the correlation (r = 0.39, p < 0.05), suggesting clustering issues may be less significant at our high-altitude site with sparse vegetation. In summary, WIBS data suggest that most ABC particles correspond to fungal spores, although pollen fragments or other plant material may also contribute.

### 3.6  DNA analysis reveals dominance of Basidiomycota in air samples

To gain a deeper understanding of fungal spore diversity beyond the identifiable (mostly) Ascomycota observed in Burkard slides, we collected 8 bi-daily air filters from September 16 to 19, 2023, extracted DNA, performed fungal barcode sequencing and taxanomic assignment. Unfortunately, due to simultaneous instrument malfunctions, no WIBS or Burkard measurements were available during the sequencing period. However, parallel impactor measurements and microscopic analysis indicate that the majority of particles with aerodynamic diameter > 1 μm during this time are likely fungal spores (see Fig. 8 for a representative example). Figure 9 displays boxplots showing the contributions of Basidiomycota and Ascomycota to the total fungal reads, revealing that the vast majority of fungal DNA originated from Basidiomycota (median value: 90.1 %). In contrast, the identifiable fungal counts from the Burkard data showed approximately 90 % Ascomycota, suggesting that the majority of unidentifiable spore counts in the Burkard data are Basidiomycota. The median contribution of Ascoymocota from the sequencing data is 8.8 %. Noticeably, 2 data points stand out: There was a snowstorm during loading of the last 2 filters (red dots in Fig. 9). For these data points, fluorescent particle concentrations (see Supplement Sect. S8) and read numbers from high-throughput sequencing were much lower. The latter is indicative of low DNA concentrations in the starting material due to low spore numbers on filters. Consequently, the absolute abundance of Ascomycota is still much lower than on the other sampling days. Within the Basidiomycota phylum, a majority of spores belong to the Agaricomycete class (median value: 99.6 %), including the 14 most abundant fungi. This finding aligns with previous studies by Maki et al. (2023); Fröhlich-Nowoisky et al. (2012); Yamamoto et al. (2012) and Fröhlich-Nowoisky et al. (2009) which reported that Agaricomycetes dominate airborne Basidiomycota and are among the most abundant atmospheric fungi. Notably, none of the Agaricomycetes identified through DNA analysis could be visually identified. Within the Agaricomycetes, the most abundant fungi belong to the order Agaricales (median: 48.7 %), Atheliales (8.0 %), Russales (4.4 %) and Polyporales (4.1 %).

Although the two methods were not used simultaneously (with a two-day gap), we compare Burkard data from shortly before the DNA sampling period (September 11 to 13, 2023) with the three days of DNA sampling under similar, although colder weather conditions (September 16 to 18, 2023). During the relevant days, 100 % of identifiable fungal spores from the Burkard data were *Cladosporium* (Ascomycota) with a mean concentration of 30.5 $\mathrm{m}^{-3}$. However, the Burkard method cannot identify all fungal spores to a taxonomic group, and it is unlikely that *Cladosporium* represents the entire fungal community. In contrast, eDNA analysis shows that *Cladosporium* accounts for only 0.44 % of all fungal reads. Assuming the eDNA data reflects the true relative abundance of fungal taxa, the measured concentration of 30.5 $\mathrm{m}^{-3}$ of *Cladosporium* spores corresponds to 0.44 % of the total fungal spore concentration. By scaling this proportion up to 100 %, we estimate the total fungal spore concentration to be approximately 6932 spores $\mathrm{m}^{-3}$. This estimate is consistent, within one order of magnitude, with the concentration of

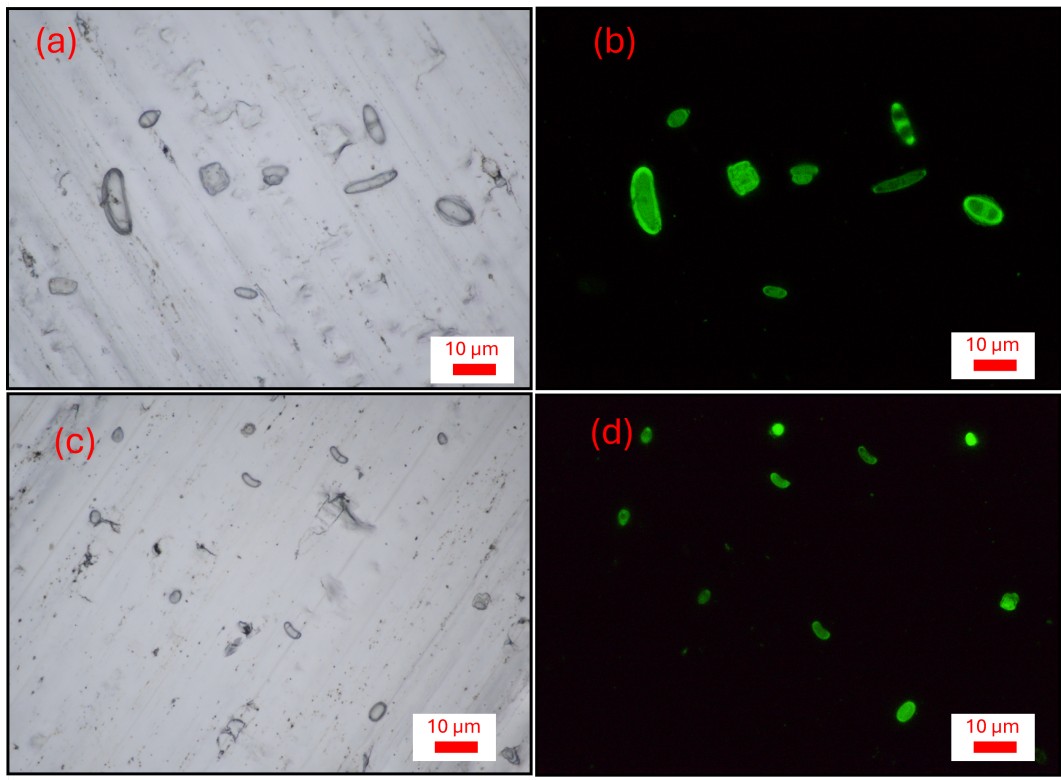

**Figure 8.** Representative examples of fluorescence and light microscopy pictures of impactor samples from September 19, 2023 9 am - 4 pm. (a) Particles from stage A (Da > 2.5 μm), (b) Same as (a) but only fluorescent particles. (c) Particles from stage B (1.0 < Da < 2.5 μm), (d) same as (c) but only fluorescent particles. Most particles show strong fluorescence and have a characteristic shape.

fluorescent particles > 1 $\mu$m aerodynamic diameter (3010 m$^{-3}$) measured with the impactor and fluorescence microscope (see Supplement Sect. S8 for details).

A large fraction of Basidiomycota uses a passive spore discharge mechanism (surface tension catapult) (Elbert et al., 2007; Pringle et al., 2005), where a drop and a film of water has to condensate on the spores in order to get expelled in the air (called ballistospores) (Zoberi, 1964). Consequently, most Basidiomycota prefer high R.H. and moderate temperatures for spore release (Fröhlich-Nowoisky et al., 2016; Gilbert and Reynolds, 2005; De Groot, 1968). Ballistospores from Basidiomycota typically range from 2 to 10 μm in diameter (Elbert et al. (2007) and references therein). As we have mostly detected reads from Basidiomycota (Fig. 9) and most of the particles we analyzed under the microscope are likely fungal spores (Fig. 8), this indicates that most fungal spores detected were basidiospores. As shown in Fig. 6 (a), the concentration of ABC particles in summer (and to a lesser extent in autumn) is strongly influenced by ambient R.H. and moderate temperatures (approximately 15 °C being most favorable). Additionally, ABC particles predominantly range between 1 and 10 $\mu$m in diameter (Fig. 10 (c)), further supporting our argument that ABC particles are primarily fungal spores of Basidiomycota.

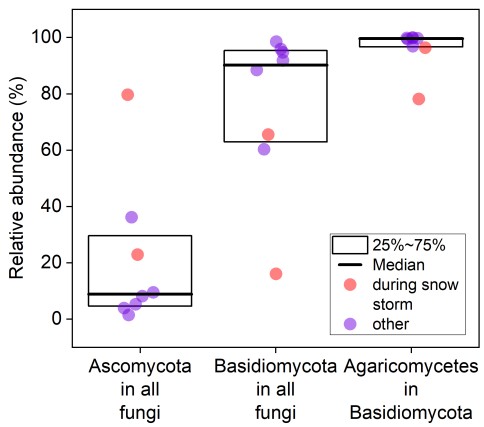

**Figure 9.** Relative abundance of Ascomycota, Basidiomycota and Agaricomycetes in Basidiomycota based on fungal reads. Data was collected from 8 individual filters.

### 3.7 Fungal spores are highly active INPs

We compared the concentration of ABC particles (and all other HFAP types) to INP concentrations over a wide range of temperature, incorporating data from PINE during the PaCe22 campaign as well as filters for INSEKT, both collected in parallel to WIBS data. We focus on ABC concentrations rather than ABC* or ABC_3, since ABC particles consist predominantly of ABC* particles and the distinction between ABC* and ABC_3 is uncommon in the existing literature, making comparisons with other studies more straightforward. For correlation analysis, PINE data was averaged to 1 h. Filters for INSEKT analysis were changed weekly, with exception for 6 bi-daily filters at the beginning and 8 bi-daily filters at the end of the second part of the campaign. The bi-daily filters were averaged for consistency with weekly filters. Figure 10 (a) shows the time series of INPs active at -13.5 °C ($N_{\mathrm{INP}}(-13.5)$) alongside ABC concentrations scaled by 0.0022 for comparison. The lowest detectable INP concentration $9 \times 10^{-4}$ L$^{-1}$ was recorded in early May, while the highest concentration (0.085 L$^{-1}$) occurred on the second week of August. Despite data points of INPs and ABC particles spanning over two orders of magnitude, all peaks and valleys compare very well. In fact, the INP-to-ABC ratio (multiplied by 0.0022) mostly remains near 1. Therefore, approximately 0.22 % of ABC particles are responsible for the INP concentration at this temperature and the contribution of ABC particles to INPs is relatively constant over the whole period. Thus, it is likely that the majority of INPs active at -13.5 °C in the immersion freezing mode are ABC particles, and therefore probably fungal spores.

The relation between $N_{\mathrm{INP}}(-13.5)$ and ABC data show the highest correlation of any WIBS-INP comparison, namely r = 0.94 (p< 0.0001) (Fig. 10 (b), marked with dashed line). In general, ABC particles gave the highest correlation coefficient for almost all temperatures. In Fig. 10 (b), the Pearson correlation coefficient for the INP-ABC correlation at all temperatures

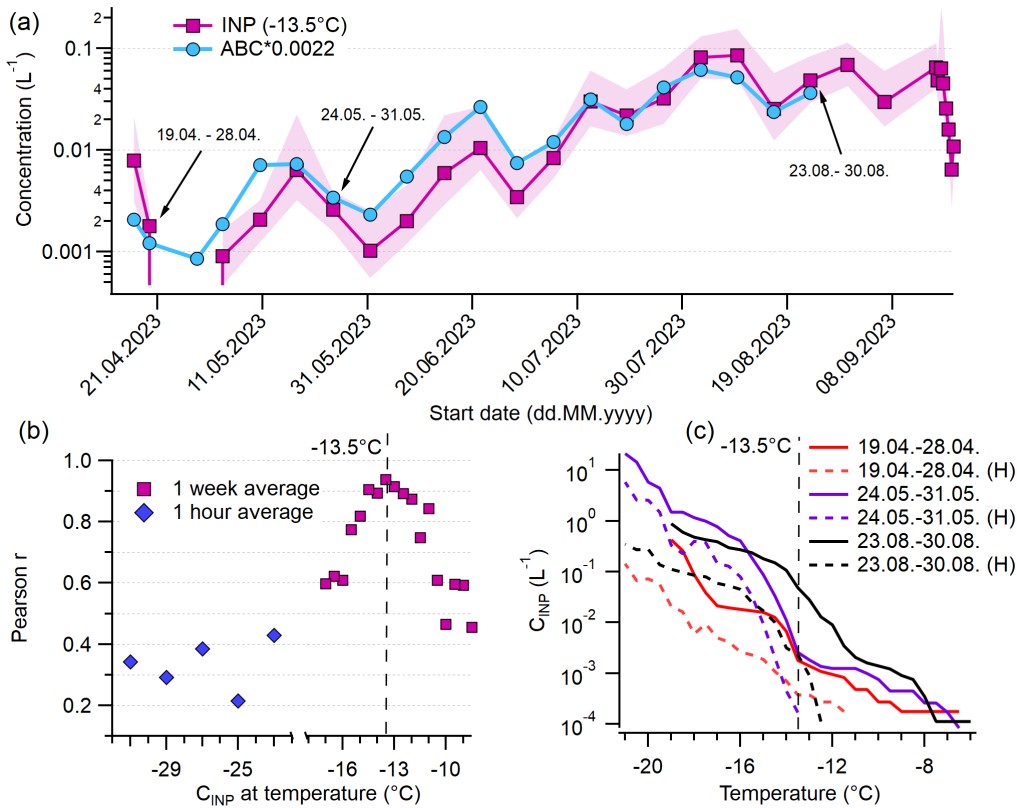

**Figure 10.** Relation of ABC particles to INPs. (a) Concentrations of weakly INPs active at -13.5 °C and ABC concentration multiplied by 0.0022. (b) Pearson correlation coefficient of correlation of ABC particles with INPs active at certain temperature. (c) Cumulative INP spectra of 3 selected days (also marked in (a)) and spectra of the same day with prior heat treatment (H). The red and black solid lines are absent in the lowest temperature range because all droplets in the original samples had frozen by –19 °C, even after a 1:100 dilution, which prevented us to measure higher INP concentrations at lower temperatures. In contrast, heat treatment reduced the INA, allowing data to extend to lower temperatures.

that reach a confidence level of 95 % are plotted. Between -12 °C and -15 °C, all Pearson values are above 0.8 (with highest p value of $1.2 \times 10^{-5}$). Hence, they are very likely to be composed of ABC particles and are therefore of fungal origin. Since AB and ABC likely share a biological source, their combined correlation was tested but showed little difference from ABC alone. Likewise, r-values for ABC* concentrations differ from those for ABC concentrations by a maximum of 0.01. At both warmer and colder temperatures, INP-ABC correlations weaken. Bacteria might play an important role in heterogeneous ice nucleation, particularly at higher temperatures (Kanji et al., 2017; Šantl Temkiv et al., 2015; Morris et al., 2004; Jayaweera and Flanagan, 1982). However, chamber studies suggest that bacteria are primarily classified as A particles in the WIBS, rather than ABC particles (Hernandez et al., 2016; Savage et al., 2017). A recent field study by Katsivela et al. (2025) at Mount

Helmos supports this, reporting a high positive correlation between total airborne bacteria concentrations and WIBS A and C concentrations (WIBS-5/NEO, $9\sigma$ threshold), with Spearman's correlation coefficients of 0.82 (p < 0.05) for both. In contrast, no significant correlation was found between bacteria concentrations and the ABC concentrations. In our study, we observed positive correlations between WIBS A concentrations and INPs active at temperatures between –9 and –15 °C, with r-values ranging from 0.70 to 0.50 (p < 0.05) (see Supplement Sect. S7, Tab. S3). This suggests that bacteria may contribute to high-temperature INPs, consistent with findings from various environments (e.g. Maki et al. (2023); Failor et al. (2017); Stopelli et al. (2017); Šantl Temkiv et al. (2015)). However, we were unable to directly quantify bacterial contributions in this study. Since particles in the A channel likely include both bacterial and non-biological components (see Supplement Sect. S2 and S3.2), the specific contribution of bacteria in high-temperature INP activity during our study remains unresolved.

High-temperature INPs are present even during snow-covered periods. In Sect. 2.3, we discuss potential local sources of ABC particles under such conditions. These include organisms that grow on trees or rocks, including lichens, mosses, and wood-decaying fungi. These organisms may also serve as sources of INPs. Recent studies by Proske et al. (2025) and Eufemio et al. (2025) demonstrate that lichens collected from diverse environments, including the Arctic and the European boreal forest, contain highly active ice nuclei that could become aerosolized. Additionally, mosses such as *Hypnum* and *Orthotrichum*, which grow preferably on tree trunks and rocks, have been shown to contain efficient ice nuclei active at temperatures above –15 °C (Moffett), suggesting they could serve as INPs. However, only their leaf material has been characterized in regard to its INA so far, not spores. Furthermore, spores from wood-decaying fungi, particularly some species within the Polyporales order, may also act as INPs. To our knowledge, however, no species or genera within this group have yet been tested for INA.

Below -15 °C, biological particles might not be the dominating source of INPs, as mineral dust, inorganic particles, and combustion aerosols also nucleate ice (DeMott and Prenni, 2010; Murray et al., 2012; Kanji et al., 2017). Pearson r-values for 1 h mean ABC and PINE data (Fig. 10 (b)) range between 0.43 (-22 °C) and 0.22 (-25 °C). AB+ABC particles had slightly higher correlations than ABC alone. r-values for HFAPs range from 0.32 to 0.47. As HFAPs most likely consist of both biological and other particles (see Supplement Sect. S2), possibly also including soil dust and mineral dust (Gao et al., 2024), the significance of biological aerosols for INPs active at colder temperatures can not be clarified. The lowest Pearson r-values for the temperatures -23 °C, -25 °C and -27 °C all belong to TAPs and are below 0.28.

For selected samples, a heat treatment with a subsequent INP analysis was performed. INP concentrations before and after heat treatment are shown for three representative filters (April, during snow cover; May, shortly after snow melt; August), marked with arrows in Figure 10 (a). After heat treatment, the vast majority of INPs active at -13.5 °C (marked with vertical dashed line) vanishes. This is typically associated with the presence of biological proteinaceous INPs whose ice nucleation ability is sensitive to heat (Daily et al., 2022; Huang et al., 2021; Šantl Temkiv et al., 2019). The concentrations of heat-labile INPs active above -13.5 °C are several orders of magnitude lower than those of ABC and total HFAP particles.

Direct comparisons between INPs and bioaerosols in field studies have grown with the adoption of LIF sensors in atmospheric science. Similar findings have been reported in a range of natural environments. For instance, Schneider et al. (2021) found that among all HFAPs, ABC particles (WIBS NEO, $9\sigma$ threshold) correlated best with INPs active at -16 °C in a boreal forest (Hyytiälä). Tobo et al. (2013) reported a significant correlation (r = 0.78, p < 0.01) between INPs at -15 °C and fluores-

cent aerosol particles (FAPs) in a Colorado Pine forest using a UV-APS in summer. Similar results were obtained by Prenni et al. (2013); Wright et al. (2014); Mason et al. (2015); Twohy et al. (2016); Cornwell et al. (2023); Kawana et al. (2024) and Taketani et al. (2025). However, these studies were not able to pinpoint the nature of the fluorescent particles. Pereira Freitas et al. (2023), combined FAP data using the multiparameter bioaerosol spectrometer (MBS, University of Hertfordshire, UK) with offline measurements of fungal spore tracers in Ny-Ålesund, Svalbad. The authors also found a high correlation of $N_{\mathrm{INP}}(-15)$ and FAPs. Additionally, the fungal spore tracers imply that the terrestrial biosphere, and potentially fungal spores are an important contributor to the INP population during arctic summer after snow melt. Size resolved measurements of INPs confirm that most INPs active at -15 °C exceed 1 μm (Da), with 70 % > 2.5 μm in the Arctic (Mason et al., 2016). This aligns with earlier studies (Mason et al., 2015; Huffman et al., 2013) which additionally found PBAPs in a similar size range, of which a majority showed typical features of fungal spores. This is in agreement with the size distribution of ABC particles measured here (Fig. 7 (c) left): The majority of ABC particles is bigger than 2.5 $\mu$m, and almost no particles < 1 $\mu$m were detected. Furthermore, Mignani et al. (2021) found that at Weissfluhjoch, Switzerland, INPs active at -15 °C were better predicted by considering only aerosol particles larger than 2 μm, rather than those larger than 0.5 μm.

The relationship between PBAPs and low-temperature INPs (< -22 °C) remains unclear. Gao et al. (2024) reported that HFAPs, measured using a WIBS-5/NEO with a $9\sigma$ threshold, contributed substantially to INP concentrations over a broad temperature range, including temperatures below –20 °C, at Mount Helmos in the eastern Mediterranean. Similar correlation coefficients of those obtained here, were reported by Vogel et al. (2024) ) with a WIBS NEO in Hyytiälä between mid-March and mid-May. However, their analysis focused on the FL3 channel (using a $9\sigma$ threshold), which encompasses ABC, BC, and C particles, rather than ABC particles specifically. In contrast Paramonov et al. (2020) found no significant correlation of fluorescent particles with low-temperature INPs, during March and April at the same location, and Lacher et al. (2021) reported that while FAPs correlated positively with INPs at Jungfraujoch (3571 m above sea level), TAPs showed an even higher correlation. In both cases, snow-covered ground likely suppressed local biogenic aerosol emissions. However, it is important to note that Paramonov et al. (2020) did not specify the fluorescence threshold used, and Lacher et al. (2021) applied a $3\sigma$ threshold. These methodological differences limit the comparability of results across studies.

## 3.8 The ice nucleation ability of detected fungal spores

In this study, we presented a comprehensive dataset of HFAPs and demonstrated, through comparisons with fungal spore counts and eDNA analysis, that the vast majority of INPs active above -15 °C are most likely fungal spores. However, as previously discussed, fungal INA has been studied much less than for example pollen INA. This is mainly due to the immense diversity of fungi (estimated at 2.2 to 5.1 million species) of which only a fraction has been discovered or described (Hawksworth and Lücking, 2017; Blackwell, 2011).

Among the fungi identified at least to the genus level via eDNA analysis (154 unique genera), species from 6 genera have previously been tested for ice nucleation. Table 3 lists these detected genera alongside the activation temperatures of specific species tested, the species identified in this study, and their mean relative abundance across the four measurement days (eight filter samples).

Not all species within a genus exhibit ice nucleation. Most tested species show very low or no ice nucleation temperature in immersion mode freezing (including all tested species of *Aspergillus*, *Trichaptum* and *Trichoderma*). It should be noted, however, that many species have only been tested at relatively warm sub-zero temperatures due to methodological limitations and may still exhibit ice nucleation at colder temperatures. Within the *Cladosporium* genus, only *C. herbarum* showed ice nucleation at -15 °C (Jayaweera and Flanagan, 1982), though this result was not confirmed by Pummer et al. (2013), likely due to differences in freezing volume (Wieland et al., 2025) or cultivation conditions (Pummer et al., 2013; Tsumuki et al., 1995). The species detected here (*C. cladosoroides*) was found to not nucleate ice (Pummer et al., 2013)). Given that *Cladosporium* counts from the Burkard trap alone should be sufficient to explain INP concentrations, it is unlikely that *Cladosporium* is responsible for the high-temperature INPs observed here. Conflicting results also exist for *Penicillium digitatum* which has been reported to nucleate ice at both -10 °C (Jayaweera and Flanagan, 1982) and -33 °C (Pummer et al., 2013). All other tested *Penicillium* species have activation temperatures of -22 °C or lower. In this study, we detected 4 species of *Penicillium* (each on one filter only) of which none has been tested for INA. The most ice active fungus known today is *Fusarium*, with a nucleation temperature around -5 °C (Kunert et al., 2019; Pummer et al., 2013; Pouleur et al., 1992). One of the four species detected here, (*F. tricinctum*) nucleates ice at -7.3 °C (Kunert et al., 2019). However, it was only detected on 1 out of the 8 filters. Two other detected species (*F. equiseti*, *F. oxysporum*) show no INA (Kunert et al., 2019; Pummer et al., 2013) and the third (*F. solani*) has not yet been tested. *Fusarium* was never detected with the Burkard trap. Thus, it remains unclear whether *Fusarium* and *Penicillium* contributes significantly to high-temperature ice nucleation. The most abundant class of fungi here are the Agaricomycetes. Some species of this class which were not detected here, have been tested for INA in laboratory studies by Haga et al. (2014) and Pummer et al. (2013) and revealed very low or no ice nucleation. However, none of the 84 unique genera of Agaricomycetes detected in this study have been tested for INA. Recent field studies suggest that Agaricomycetes may contribute to atmospheric high-temperature ice nucleation (Tang et al., 2022). Specifically *Hypholoma*, one of the most abundant fungi here (median relative abundance of 3.4 % (mean: 4.3 %)) as well as *Sistotrema* (median: 1.1 % (mean: 5.3 %)), have been proposed as high-temperature ice nuclei, based on comparison of relative atmospheric abundance and INP concentrations (Niu et al., 2024).

## 4 Conclusions

This study presents size-resolved measurements of HFAPs over 4 season in the Finnish sub-Arctic, demonstrating that AB and ABC particles dominate the HFAP fraction during warmer periods. Meteorological conditions, particularly snow cover, significantly influence AB and ABC particle concentrations. Snow cover led to an immediate reduction in these particle types, suggesting that as soon as the local biosphere is (partially) buried under a thick layer of snow, the emission of PBAPs is reduced strongly. Diurnal pattern confirmed the local origin of the PBAPs, as concentrations where highest when the station was inside the mixing layer, and therefore coupled to the local surface. A comparison with direct fungal spore and pollen counts confirmed that ABC particles predominantly consist of fungal spores, and a newly applied fluorescence threshold strategy revealed that a minor fraction of ABC particles correlated with pollen grains. Visual identification of fungal spores using the

**Table 3.** Ice nucleation temperatures (immersion freezing) of species whose genus was detected via eDNA analysis on filter samples and the detected species and its mean relative abundance. The number in brackets refers to the number of filters the species was detected on (out of 8 filters) $T_{50}$ and $T_{96}$ are the temperatures at which 50 % and 96 % of sample containing droplets froze, respectively, and $T_{onset}$ is the temperature at which the first droplet froze. Species in bold letters are the only species detected, which have been tested for INA before. *sp.* means that different species of the genus are grouped together because they cannot be resolved more precisely.

| Genus | Species detected | mean abundance (%) | Species | INA (°C) | Reference |
|---|---|---|---|---|---|
| *Cladosporium* | **C. cladosoroides** | 0.38 (5) | **C. cladosoroides** | no INA | (Pummer et al., 2013), $T_{50}$ |
| | | | C. herbarum | no INA | |
| | | | C. herbarum | -15 | (Jayaweera and Flanagan, 1982), $T_{96}$ |
| | | | C. sp. | -35.1 | (Iannone et al., 2011), $T_{50}$ |
| *Penicillium* | P. bialowiezense | 0.13 (1) | P. notatum | -22 | |
| | P. melinii | 0.07 (1) | P. frequentes | -22 | (Jayaweera and Flanagan, 1982), $T_{96}$ |
| | P. rubens | 0.04 (1) | P. digitatum | -10 | |
| | | | P. digitatum | -33 | (Pummer et al., 2013), $T_{50}$ |
| | | | P. chrysogenum | no INA | |
| | | | P. citrinum | no INA | |
| | | | P. glabrum | no INA | |
| | | | P. sp | -27 | (Haga et al., 2014), $T_{onset}$ |
| | | | P. brevicompactum | -27 | |
| *Aspergillus* | A. penicillioides | 0.29 (4) | A. brasiliensis | -25 | |
| | A. tardicrescens | 0.02 (1) | A. niger | -29 | (Pummer et al., 2013), $T_{50}$ |
| | | | A. niger | no INA | |
| | | | A. fumigatus | -34 | |
| | | | A. oryzae | -31 | |
| *Fusarium* | **F. tricinctum** | 0.13 (1) | **F. tricinctum** | -7.3 | (Kunert et al., 2019), $T_{50}$ |
| | **F. equiseti** | 3.8 (3) | **F. equiseti** | no INA | |
| | **F. oxysporum** | 0.04 (1) | **F. oxysporum** | no INA | |
| | F. solani | 0.02 (1) | **F. oxysporum** | no INA | (Pummer et al., 2013), $T_{50}$ |
| | | | F. acuminatum | -3.5 to -6.3 | (Kunert et al., 2019), $T_{50}$ |
| | | | F. acuminatum | -5 | (Pouleur et al., 1992), $T_{onset}$ |
| | | | F. avenaceum | -2.5 | |
| | | | F. avenaceum | -9 | (Pummer et al., 2013), $T_{50}$ |
| | | | F. avenaceum | -5.0 to -7.6 | (Kunert et al., 2019), $T_{50}$ |
| | | | F. armeniacum | -5.3 | |
| | | | F. begoniae | -11.2 | |
| | | | F. concenticum | -4.6 | |
| | | | F. langsethiae | -7.3 | |
| *Trichaptum* | T. fuscoviolaceum | 0.11 (2) | T. abietnium | -26 | (Haga et al., 2014), $T_{onset}$ |
| *Trichoderma* | T. rossicum | 0.04 (1) | T. atroviride | -32 to -35 | (Pummer et al., 2013), $T_{50}$ |
| | T. aeroaquaticum | 0.02 (1) | T. longibrachiatum | -32 | |
| | T. harzianum | 0.02 (1) | T. reesei | no INA | |
| | | | T. virens | no INA | |
| *Rhizopus* | R. microsporus | 0.02 (1) | R. stolonifera | -23 | (Jayaweera and Flanagan, 1982), $T_{96}$ |

Burkard trap is limited to a taxonomic group where reliable recognition is possible, and accounted for only a fraction (3–10 %) of total fungal spores. This finding was supported by eDNA analysis, which showed that most fungal genera detected belong to Basidiomycota, particularly Agaricomycetes, of which none could be visually identified as such. A strong correlation between ABC particle concentrations and high-temperature INPs suggests that fungal spores are the dominant INPs active at temperatures around -13.5 °C. However, eDNA analysis did not pinpoint specific ice-nucleating fungal species, as only a small subset of fungal genera has been tested for ice nucleation activity, and activation temperatures above -15 °C are rare among tested genera. This leaves an open question regarding the exact biological particles responsible for high-temperature ice nucleation. Despite this uncertainty, our study marks a significant step in demonstrating that fungal spores, likely originating from the local biosphere, dominate the INP population active above -15 °C. Further testing of Agaricomycetes for INA is needed to determine which specific species or genera contribute to atmospheric ice nucleation. However, Basidiomycota (which include Agaricomycetes) are generally more difficult to cultivate than Ascomycota (Rungjindamai and Jones, 2024; Krupodorova et al., 2021). Additionally, cultivation conditions and strain selection can influence INA (Pummer et al., 2013; Tsumuki et al., 1995), posing a significant challenge for future research.

The fungal fruiting season is expected to lengthen due to climate change (Boddy et al., 2014), and the fungal spore season is advancing (Wu et al., 2024). Simultaneously, the snow-covered period in the subarctic is projected to shorten (Dankers and Christensen, 2005). These changes suggest that fungal ice nucleation in Arctic and sub-Arctic regions may become increasingly significant, as ice-active fungal spores could remain available over a longer portion of the year. Future climate models should account for these shifts to better understand the impact of fungal high-temperature INPs on low-level cloud feedback at high latitudes.

*Data availability.* WIBS data from the PaCE22 campaign is available at the open data repository Zenodo under the doi 10.5281/zenodo.13885888 (Gratzl, 2024). A data description paper is provided by Gratzl et al. (2025).

WIBS data from the second part of the campaign and relative contributions of fungal species via eDNA analysis is available at the open data repository TU Wien Research Data under the doi 10.48436/0xmsb-eps20 (Gratzl and Grothe, 2025).

PINE data from the PaCE22 campaign is available at the open data repository Zenodo under the doi 10.5281/zenodo.13889647 (Böhmländer et al., 2024). A data description paper is provided by Böhmländer et al. (2025).

INSEKT data is available at the open data repository radar4KIT under the doi 10.35097/bb9t8cv9gfaghcdc (Böhmländer, 2025).

Burkard data will be available in the context of the SYLVA (A SYstem for ReaL-Time ObserVation of Aeroallergens) project in near future or upon request.

Meteorological data, eBC, SO$_2$ and NOx concentrations are available at https://en.ilmatieteenlaitos.fi/download-observations (last access: 26.02.2025) for the observation station "Muonio Sammaltunturi".

*Author contributions.* J.G., A.B., D.B., K.D., E.A., O.M. and H.G. conceived and planned the experiments. D.B. and K.D. prepared and organized the PaCE 2022 campaign. J.G. performed WIBS measurements and data analysis, analyzed impactor foils with fluorescence

microscopy, combined and interpreted all data with contributions from all authors, visualized data. J.G., F.W., and O.M. collected filter samples. A.B. performed PINE and INSEKT measurements and analyzed data. S.P. and C.P. performed pollen and spore counts and analyzed data. C.P. and M.G. performed the eDNA measurements and analyzed data. J.G. wrote the manuscript with support from A.B., S.P., C.P., M.G., K.D. and E.A. All authors discussed the results and reviewed and edited the manuscript.

*Competing interests.*  At least one of the (co-)authors is a member of the editorial board of Atmospheric Chemistry and Physics.

*Acknowledgements.*  This work was supported by atmo access under the ID ATMO-TNA-4-0000000069. We gratefully acknowledge support by the FFG (Austrian Research Promotion Agency) for funding under project no. 888109, the Vienna Science and Technology Fund (WWTF) through project VRG22-003, and the Research Council of Finland for ACCC Flagship (project no 359342). The authors gratefully acknowledge the NOAA Air Resources Laboratory (ARL) for the provision of the HYSPLIT transport and dispersion model and/or READY website (https://www.ready.noaa.gov) used in this publication. We thank TU Wien and BMBWF for the purchase of the WIBS 5/NEO. The authors would like to thank the team from Metsähallitus and Sammaltunturi station for their help with logistics, filter changing and watch over our instrumentation. The authors acknowledge the TU Wien Bibliothek for financial support through its Open Access Funding Program. We are also grateful to Anne Kasper-Giebl from TU Wien and Paul Winkler from the University of Vienna for the equipment they lent us. We thank Tobias Könemann from Droplet Measurement Technologies for his technical support. We thank Harald Berger (Symbiocyte, Vienna, Austria) who did the initial bioinformatics. We also like to thank Evgeny Kadantsev from FMI for the fruitfull dicussions.

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
