# Peer review of "Locally emitted fungal spores serve as high-temperature ice nucleating particles in the European sub-Arctic"

_EGUsphere, 2025_

## Author Response (AR1)

We thank all Referees for their valuable comments and suggestions to improve our manuscript. Please find below the Referee's comments in black, our answers in blue and parts of the manuscript in red, with changes and additions underlined.

**Referee 1:**

This study presents a comprehensive analysis to show that locally emitted fungal spores can serve as ice nucleation particles at high temperatures at a site in the European sub-Arctic. The analysis of the fluorescent aerosols is in-depth and convincing. This study provides diverse methods to consolidate the statements, including chemical, biological, and meteorological techniques. I believe that it is worth publishing in this journal when the following issues are resolved. Overall, I suggest a major revision based on the comments below.

Major comments:

1. 3.3: When the station is in-cloud, how can cloud-processing can influence the concentration and type of the HFAPs, especially AB and ABC particles?

We thank Referee 1 for raising this important question. We have already dedicated a whole section to this issue in the Supplement (Sect. S4: Inside cloud and precipitation) where we discuss the changes in concentration when the station is inside clouds (and during rain events) and possible reasons for differences in TAP and HFAP concentration changes. In summery: During the measurement period, the station was immersed in clouds 27 % of the time, with the highest occurrence in autumn and winter. Cloud immersion significantly reduced total aerosol particle (TAP) and highly fluorescent particle (HFAP) concentrations due to cloud and rain scavenging. TAPs declined more strongly, leading to a marked increase in the fluorescent fraction, especially for ABC-type particles. This suggests that bioaerosols are less efficiently removed in clouds, likely because they are poor CCNs and remain in the interstitial phase.

We reference this Supplement Sect. both in Sect. 3.3 (line 299) and 3.4. (line 330). However, we understand that these references to the Supplementary Materials may be easy to overlook, as they currently appear at somewhat unexpected points in the manuscript. We appreciate Referee 1 for bringing this to our attention. We therefore extended the sentence at line 330 from *"The effect of rain is discussed in the Supplement in Sect. S4."* to

Several studies have reported increases in bioaerosol concentrations during and after rain events (Rathnayake et al., 2017; Yue et al., 2016; Gosselin et al., 2016; Heo et al., 2014; Schumacher et al., 2013; Huffman et al., 2013; Prenni et al., 2013; Allitt et al., 2000; Hist and Stedman, 1963; Gregory and Hirst, 1957). In contrast, no such relationship was observed in our study between rainfall and increased concentrations of HFAPs. A likely explanation is that, during most rain events recorded at our station, the site was simultaneously within a cloud. These in-cloud conditions led to a marked decrease in the concentrations of both TAPs and HFAPs due to cloud and rain scavenging (Flossmann and Wobrock, 2010; Sellegri et al., 2003). However, HFAPs, particularly the AB and ABC subtypes, showed a much smaller decline compared to TAPs. This resulted in a substantial relative increase in the AB and ABC fractions during in-cloud events. Further details on the effect of rain and cloud immersion on the concentrations and fractions of HFAPs and TAPs are discussed in the Supplement Sect. S4.

Furthermore, we changed the paragraph in the Supplement (line 93 – 96), which is now discussed in the main text to respond to Referee 3, comment 18:

Figure S9 also displays rain intensity for the selected periods. As discussed in Sect. 4 of the main manuscript, we generally did not observe an increase in HFAP concentrations during or after rainfall. Notably, during most rain events shown, the station was also immersed in cloud, suggesting that cloud and rain scavenging suppressed aerosol concentrations. This likely outweighs the rain-related increases in bioaerosols reported by previous studies (Rathnayake et al., 2017; Yue et al., 2016; Gosselin et al., 2016; Heo et al., 2014; Schumacher et al., 2013; Huffman et al., 2013; Prenni et al., 2013; Allitt et al., 2000; Hirst and Stedman, 1963; Gregory and Hirst, 1957). To provide context for the examples in Fig. S9, rain occurred during approximately 23 % of the snow free measurement period. The duration and intensity of rain events varied greatly ranging from brief light showers to multi-day episodes of sustained precipitation. The two case studies shown represent different types of rainfall patterns commonly encountered during the campaign.

2. 3.4 and Figure 6: please explain why in different seasons, the patterns are different for the same meteorological factor.

We thank the referee for pointing out the seasonal differences in the influence of meteorological factors on ABC particle concentrations. These variations likely reflect seasonal changes in biological activity and surface conditions. In response to this valuable comment, as well as a similar suggestion from Referee 3, we have expanded the discussion in this section to provide a more detailed interpretation. Additionally, we have modified the y-axis of Fig. 6 to a logarithmic scale to better visualize changes in ABC concentrations across different meteorological conditions. The revised paragraph now reads:

In spring and winter, R.H. has little effect on ABC particle concentration, whereas in summer, a positive relationship appears, with maximum median concentrations occurring at 80-100 % R.H. (Fig. 6 (a)). A similar but weaker trend is observed in autumn. In spring, summer and autumn, ABC concentrations increase with temperature up to 15 °C but decline at higher temperatures in summer (Fig.6 (b)). Wind speed has an effect in summer, with peak median concentrations on calm days (< 4 m s$^{-1}$), while in winter, there is a slight positive correlation between ABC concentration and increasing wind speed (Fig. 6 (c)).

The maximum concentrations at high R.H. and the increase with temperature up to approximately 15 °C with a decline at higher temperatures, may reflect a fungal spore release mechanism (e.g. surface tension catapult; Pringle et al., 2005) which favors high humidity and moderate temperatures. The high fungal activity in autumn could also be responsible for the higher ABC concentrations observed already at lower temperatures compared to summer. In the same sense, in spring, fungal activity is lower, resulting in almost no R.H. dependence. Notably, the highest concentrations in spring are observed when the surrounding ground is still snow-covered (see Fig. 3), and when air masses originate from the south (see Supplement Sect. S3.3), where snowmelt has already occurred. This suggests that PBAPs may be transported to Pallas with these warmer air masses (and often lower R.H.). In winter, most local sources are buried under snow and no trend with R.H. and temperature is observed at the overall low concentrations of ABC particles. However, the slight positive trend between ABC concentrations and wind speed suggests that the few potential local sources such as cellulolytic fungi, lichens, or mosses (see Sect. 3.2), may be more efficiently aerosolized under stronger winds (Ščevková et al., 2024; Armstrong, 1991; Kallio, 1970), transported to the site, or result from biological residues in blowing snow (Boetius et al., 2015).

In summary, ABC concentrations peak under high R.H., moderate temperatures (10–15 °C) and low wind speeds in summer. In autumn, both temperature and RH positively influence concentrations, although wind speed plays a lesser role. In spring, temperature (up to 15 °C) is the dominant factor, while in winter, wind speed is the only meteorological variable with a clear impact. Similar seasonal patterns are observed for AB particles (see Supplement Sect. S9), whereas B and BC particles show no clear relationship with R.H. but tend to increase with temperature in both summer and spring.

[Figure]

**Figure 6.** The median concentration of ABC particles dependent on meteorological parameters for each season. Depicted are median concentration values for 1 h mean values. (a) R.H., (b) ambient temperature, (c) wind speed.

3. Section 3.7: in the last two paragraphs of this section, the authors discuss the observations of INPs and fluorescent aerosols in other studies. However, most of discussions do not differentiate highly fluorescent aerosols and all fluorescent aerosols. This might be partly the reason that some observations do not find relationships. Please refine the discussions by explicitly considering the fluorescence intensity.

We agree that this is a very important point made. We added both the WIBS-model and the threshold used to the discussion. In line 509, it now reads:

For instance, Schneider et al. (2021) found that among all HFAPs, ABC particles (WIBS NEO, 9σ threshold) correlated best with INPs active at -16 °C in a boreal forest (Hyytiälä).

Further discussion of correlations between high temperature INPs and fluorescent aerosol particles primarily draws on two studies using a UV-APS and a multiparameter bioaerosol spectrometer. As these instruments differ in threshold settings, a detailed explanation of their threshold criteria would be beyond the scope of this manuscript.

In lines 524–529, we discuss correlations of WIBS data with low temperature INPs, and extended the discussion as follows:

The relationship between PBAPs and low temperature INPs (< -22 °C) remains unclear. Gao et al. (2024), reported that HFAPs, measured using a WIBS-5/NEO with a 9σ threshold, contributed substantially to INP concentrations over a broad temperature range, including temperatures below –20 °C, at Mount Helmos in the eastern Mediterranean. Similar correlation coefficients of those obtained here, were reported by Vogel et al. (2024) with a WIBS NEO in Hyytiälä between mid-March and mid-May. However, their analysis focused on the FL3 channel (using a 9σ threshold), which encompasses ABC, BC, and C particles, rather than ABC particles specifically. In contrast Paramonov et al. (2020) found no significant correlation of fluorescent particles with colder temperature INPs, during March and April at the same location and Lacher et al. (2021) reported that while FAPs correlated positively with INPs at Jungfraujoch (3571 m above sea level), TAPs showed an even higher correlation. In both cases, snow-covered ground likely suppressed

local biogenic aerosol emissions. However, it is important to note that Paramonov et al. (2020) did not specify the fluorescence threshold used, and Lacher et al. (2021) applied a 3σ threshold. These methodological differences limit the comparability of results across studies.

Specific comments:

1. L27-29: currently, the link between the first part of this sentence and the "their activation temperature" is not clear. Please rephrase this sentence.

We rephrased it to

Model studies suggest that the radiative properties of low-level Arctic clouds are sensitive to variations in INP concentrations (Gjelsvik et al., 2025; Xie et al., 2013). This indicates that the abundance of INPs and the temperatures at which they become active may be a crucial factor in understanding Arctic amplification (Murray et al., 2021; Tan et al., 2022).

2. L162: "identifiable spore if both are meant": not clear. Do you mean impossible to distinguish between these two types?

Identifiable spores" refers to the sum of identifiable fungal spores and identifiable plant spores, although they can be distinguished. We used this term, as the correlation of ABC concentrations is highest when both plant and fungal spore concentrations are added together. We changed *"or identifiable spore if both are meant."* to

The sum of both identifiable fungal spores and identifiable plant spores are referred to as identifiable spores.

3. Figure 2a: I suggest modifying the axis label as "Fraction of HFAPs in TAPs".

Done!

4. Figure 2 legend: I suggest adding one sentence to tell the reason why AC is not shown in the plot.

We added the following sentence at the end of the caption:

The fraction of AC particles is not shown, as they contributed only 0.01 % or less to HFAP concentrations.

5. Figure 3: the y axis for temperature is absent. "Horizontal transparent red lines" should be "Vertical ...".

We added the y-axis for temperature, changed "*horizontal*" to "*vertical*" and deleted "certain" in the caption. Fig. 3 is now

[Figure]

**Figure 3**. Daily mean concentrations of TAPs and HFAPs, as well as AB and ABC particle types, mean snow depth and ambient temperature inside a near spruce forest. (a) Temperature measured at Kenttärova station is used to define the meteorological seasons. Grey lines are at 0 °C and 10 °C. (b) TAPs and HFAPs, (c) AB and ABC particles. Vertical transparent red lines are labeled with a number and mark the two days discussed in the text. Snow cover reduces the concentration of AB and ABC particles drastically.

6. Figure 4a: what does "global" mean in "global radiation"?

Global radiation is the total radiation onto a horizontal surface by both direct radiation by the sun and diffuse radiation and a standardized term in meteorology which, in our opinion, does not need further explanation.

7. Figure 4b: Do NON-HFAP also include FAPs other than the HFAPs?

NON-HFAPs also include the particles that are classified as FAPs but not as HFAPs. To clarify this, we added the following sentence to the caption of Fig. 4 (b) after *"(b) Concentration of NON-HFAPs and HFAPS.":*

NON-HFAPs can also contain FAPs that did not classify as HFAPs.

8. L347: please add the data for AB, e.g., in SI.

We added the following graph (with log y-axis, since we also changed the axis in Fig. 6) and added a new Section S9 to the Supplement, reading:

9 AB concentrations in relation to meteorology

The relationship of median AB concentrations for each season is shown in Fig. S16 and reveals strong similarities to ABC concentrations.

[Figure]

**Figure S16:** The median concentration of AB particles dependent on meteorological parameters for each season. Depicted are median concentration values for 1 h mean values. (a) R.H., (b) ambient Temperature, (c) wind speed.

9. Figure 6: in (b) the ABC concentration can > 5 *10-3 cm-3, however, in (c) the same overall data become <2 when binned by wind speed. Please check the processing of the data.

We thank Referee 1 for this note. We have carefully re-checked the data processing and confirmed that everything is in order. In Fig. 6, the data are displayed as median values within each bin. As individual high values are more randomly distributed across the wind speed bins (e.g. there is no clear relationship between ABC concentration and wind speed in spring), they have less influence on the median. In contrast, the temperature bins show a clearer accumulation of higher values at elevated temperatures, which has a stronger effect on the median. Adding the sum over all bins, instead of the median values, gives the same result for both temperature and wind speed.

10. L480: not only specific to "this temperature", but also to the characterized freezing mode.

We changed the sentence, also regarding Referee 3's suggestion to use a softer statement, to

Thus, it is likely that the majority of INPs active at -13.5 °C in the immersion freezing mode are ABC particles, and therefore probably fungal spores.

11. Figure 10c: why are the parts for red and black solid lines at the lowest T range absent while those of the dashed lines are there?

The reason is that in the original samples, even after a 1:100 dilution of the suspension, all droplets had already frozen by –19 °C. After heat treatment, fewer droplets froze at higher temperatures, enabling us to obtain data at lower activation temperatures.

We added the following to the figure caption:

The red and black solid lines are absent from the lowest temperature range because all droplets in the original samples had frozen by –19 °C, even after dilution by a factor 100.In contrast, heat treatment reduced the INA, enabling the data to be extended to lower temperatures.

12. Figure S3: (a) and (b) are the same as those in Figure 4. Just keep (c).

We changed the figure regarding your suggestions and have adapted the caption accordingly:

[Figure]

**Figure S3**. Diurnal cycle of summertime concentrations of A, B and BC particles. Diurnal trends of in-cloud frequency, ambient temperature, TAPs and other HFAP types are depicted in the main manuscript in Fig. 4.

13. SI text 1: both WIBS and APS can characterize the size distribution of the aerosols. A direct comparison between the size distributions of total aerosol particles is much better than the overall comparison of the total concentration in reflecting the characterizing ability of the two instruments.

We agree that the comparison of the size distributions is valuable. We added the size distributions to Fig. S1 and extended Sect.S1 of the Supplement to

[Figure]

Figure S1: (a) Scatter plot of TAP concentrations measured with WIBS vs. concentrations of particles with aerodynamic diameter > 523 nm measured with APS with 30 min resolution. The black line represents the 1:1 line. (b) comparison of arithmetic mean size distributions during the PaCE22 campaign of TAPs of WIBS and APS and their ratio.

An Aerodynamic Particle Sizer (TSI, USA) APS was operated alongside WIBS to compare particle concentrations and validate WIBS data. The APS measures particles with an aerodynamic diameter > 0.523 µm. 30 min mean values of APS data and total aerosol particles (TAPs) from WIBS were calculated and compared during the PaCE22 campaign. APS concentrations were on average 3 times higher than those recorded by WIBS. A scatter plot of TAP concentration measured with WIBS as a function of the TAP concentration measured with APS can be seen in Fig. S1 (a). Average size distributions of TAPs measured by both the WIBS and APS during the entire PaCE22 campaign, as well as their ratio are shown in Fig. S1 (b). It needs to be noted that both devices use different approaches for size-classification (optical versus aerodynamic) and are hence not directly comparable. On average, WIBS recorded higher concentrations of particles in the 3–7 µm size range, whereas concentrations of particles smaller than 3 µm and larger than 7 µm were lower than those measured by the APS. The lower concentrations of larger particles detected by WIBS are likely due to particle losses in its inlet system (Gratzl et al., 2025). Additionally, because the aerodynamic diameter (measured by APS) tends to be larger than the optical equivalent diameter (measured by WIBS), APS may have registered smaller particles as

belonging to larger size bins. This would contribute to higher concentrations at smaller sizes compared to WIBS.

Technical comments:

1. L101: to me, "LIF" is more commonly used than "LiF"

We changed all abbreviations to "*LIF*".

2. L175-176: "AE", "LGC", and "ITS" are not defined.

3. L189: "LMP" is not defined.

We defined all these abbreviations from comment 2 and 3. Please refer to our answer to Referee 2, comment 20.

4. L255: "does no" should be "does not".

We corrected it.

5. "3.1 The contributions of different HFAPs changes seasonally": "changes" should be "change"

Thank you! We changed it.

6. L549: Should be "of which none has been tested for INA".

We changed "non" to "none".

7. L552: "two" should be capitalized for the first letter.

Done!

8. L559: should be "one of the most abundant".

We corrected it accordingly.

**Referee 2:**

Gratzl et al. present findings from a long-term monitoring campaign using a plethora of aerosol, bioaerosol and INP measurement instrumentation to determine the identity of high-temperature ice-nucleating particles as being local fungal spores in the European sub-Arctic. The analysis and correlations from the multiple datasets are robust and in-depth, honing in on the identification of fungal spore populations as defining the INP population and the influence that this finding could have in a warming climate. The work represents a step forward in better understanding the nature of biological INPs in the atmosphere, which is notoriously difficult given the complexity of such samples, and highlights the need for long-term monitoring campaigns with appropriate instrumentation and design of experiments for elucidation of atmospheric INP populations. This article demonstrates very useful findings and I believe it should be published following minor revisions, in particular one major comment surrounding bacterial INPs.

Major comments:

1. Fungal spores and pollen are discussed in detail, but I am surprised that there is no discussion on the potential influence of bacteria. 16S analysis is often performed alongside ITS. It is surprising to not see the influence of bacteria (and their proteins) investigated despite their mention in the Introduction and a brief discussion at the end of the article. The authors give the impression that bacterial INPs are active at much warmer temperatures (> -10 °C), but this is not necessarily always the case. Table 2 makes clear that fungal spores can cover a range of activities from -3.5 °C to -35 °C (or have no INP activity), but the same argument is not made here for bacteria, and so bacterial influences are dismissed despite being in a rich biome that is likely home to a wide range of bacterial diversity.

We thank Referee 2 for this important comment. Our decision to focus exclusively on ITS2 sequencing was based on a cost-benefit analysis. High-throughput sequencing is resource-intensive and given our findings indicating that ABC particles are both INPs and fungal spores, we prioritized fungal DNA analysis. However, we fully acknowledge that bacteria may also contribute to ice nucleation within the temperature range discussed. Therefore, we have expanded the discussion on bacterial contributions in line 489 as follows:

Bacteria might play an important role in heterogeneous ice nucleation, particularly at higher temperatures (Kanji et al., 2017; Šantl Temkiv et al., 2015; Morris et al., 2004; Jayaweera and Flanagan, 1982). However, chamber studies suggest that bacteria are primarily classified as A particles in the WIBS, rather than ABC particles (Hernandez et al., 2016; Savage et al., 2017). A recent field study by Katsivela et al. (2025) at Mount Helmos supports this, reporting a high positive correlation between total airborne bacteria concentrations and WIBS A and C concentrations (WIBS-5/NEO, 9σ threshold), with Spearman's correlation coefficients of 0.82 (p ≤ 0.05) for both. In contrast, no significant correlation was found between bacteria concentrations and ABC concentrations. In our study, we observed positive correlations between WIBS A concentrations and INPs active at temperatures between –9 and –15 °C, with r-values ranging from 0.70 to 0.50 (p < 0.05) (see Supplement Sect. S7, Tab. S3). This suggests that bacteria may contribute to high temperature INPs, consistent with findings from various environments (e.g. Maki et al., 2023; Failor et al., 2017; Stopelli et al., 2017; Šantl Temkiv et al., 2015). However, we were unable to directly quantify bacterial contributions in this study. Since particles in the A channel likely include both bacterial and non-biological components (see Supplement Sect. S2 and S3.2), the specific contribution of bacteria in high temperature INP activity during our study remains unresolved.

As in the meantime, new research appeared in the connection of WIBS A particles and bacteria (the above discussed publication by Katsivela et al. (2025)), we changed the sentence *"In chamber experiments, bacteria have been classified almost exclusively as A particles (Savage et al., 2017; Hernandez et al., 2016), although field studies have not confirmed this correlation."* In line 23 of the Supplement to

In chamber experiments, bacteria have been classified almost exclusively as A particles (Savage et al., 2017; Hernandez et al., 2016), although only a single field study confirmed this positive correlation (Katsivela et al., 2025).

In response to Referee 4, who pointed out that HFAPs and INPs are still present during the snow-covered period, we extended the discussion to address lichen, mosses, and tree decaying fungi as possible winter sources of INPs. It reads:

High temperature INPs are present even during snow-covered periods. In Sect. 2.3, we discuss potential local sources of ABC particles under such conditions. These include organisms that grow on trees or rocks, including lichens, mosses, and wood-decaying fungi. These organisms may also serve as sources of INPs. Recent studies by Proske et al. (2025) and Eufermio et al. (2025) have demonstrated that lichens collected from various environments, including the Arctic and the European boreal forest, contain highly active ice nuclei that could become aerosolized. Additionally, mosses such as *Hypnum* and *Orthotrichum*, which grow preferably on tree trunks and rocks, have been shown to contain efficient ice nuclei active at temperatures above –15 °C (Moffett, 2015), suggesting they could serve as INPs. However, only leaf material has been characterized in regard to its INA so far, not spores. Furthermore, spores from wood-decaying fungi, particularly some species within the *Polyporales order*, may also act as INPs. To our knowledge, however, no species or genera within this group have yet been tested for INA.

2. Line 455: Make it clear that Cladosporium is an Ascomycota. This is somewhat confusing, however, as on one hand Cladosporium is stated as being the most abundant identifiable fungal spore type, but then Ascomycota are shown in Fig 9 as being low in relative abundance. Were identifiable spore types in very low abundance compared to the other classifications?

Cladosporium was the most abundant identifiable fungus in the Burkard data, as it could be recognized through visual inspection. Figure 9 shows the relative abundance of Ascomycota based on eDNA sequencing, underscoring the discrepancy between what can be visually identified and the broader fungal diversity present in the atmosphere. This suggests that relying solely on morphologically identifiable spores significantly underrepresents the actual airborne fungal population. We conclude that visually identifiable spores constitute only a small fraction of the total fungal spore population. This is further supported by Tab. 1, which demonstrates that identifiable fungal spores account for only a subset of the total number of spores (including those confidently visually identified as fungal but lacking taxonomic resolution). We revised the sentence at line 455 to:

During the relevant days, 100 % of identifiable fungal spores from the Burkard data were *Cladosporium* (Ascomycota) (...)

Please also refer to our answer to comment 35, where this is addressed again.

Minor comments:

1. Line 53: Some additional fungal INPs may be found in the recent list compiled by Tarn et al.: https://doi.org/10.1063/5.0236911

We added Tarn et al. (2025) to the references in line 53–54.

2. Line 60: The authors should cite recent papers from the group of Santl-Temkiv related to biological sources of INPs in the Arctic; while these are related to INPs in marine waters, they may represent an atmospheric source: https://doi.org/10.5194/ar-3-81-2025; https://doi.org/10.5194/acp-25-3327-2025

Thank you, we acknowledge that these important papers do fit well for citations at that point. We added Jensen et al., 2025 and Wieber et al., 2025 after the sentence in line 60.

3. Line 61: What about the Arctic Haze?

We thank Referee 2 for this comment. Arctic haze is indeed an important seasonal contributor to aerosols in the Arctic and sub-Arctic during winter and early spring, primarily characterized by long range transported anthropogenic pollution. However, Arctic haze is generally associated with sulfates, black carbon and particulate organic matter. In contrast, the high temperature INPs discussed here exhibit a distinct summer peak, which is more likely driven by local or regional biological sources. For clarification, we have expanded the paragraph, in which we also address Referee 3's comment 9:

Recent studies have identified terrestrial environments as important, though not exclusive, sources of biological high temperature INPs (active above –15 °C) in the Arctic (Jensen et al., 2025; Wieber et al., 2025; Tobo et al., 2024; Pereira Freitas et al., 2023; Conen et al., 2016). In northern latitudes, INPs active above –15 °C show a distinct seasonal pattern, with concentrations peaking in summer during snow- and ice-free periods (Barry et al., 2025; Tobo et al., 2024; Pereira Freitas et al., 2023; Schneider et al., 2021; Wex et al., 2019). Fu et al. (2013) and Pereira Freitas et al. (2023) observed similar seasonal cycles in arabitol and mannitol, chemical tracers of fungal spores (Bauer et al., 2008), which they also attributed to terrestrial sources, in contrast to Arctic haze which dominates the aerosol composition during winter and early spring (Beck et al., 2024; Asmi et al., 2011; Quinn et al., 2007).

4. Line 64: "Attributed" instead of "contributed".

We changed it to attributed.

5. Line 85: Does this campaign (particularly the second half in 2023) have a name?

No.

6. Line 88: Give the size range of the APS.

We added the size range in brackets:

Aerodynamic Particle Sizer APS 3321 (TSI, USA, aerodynamic diameter 0.5 – 20 μm) (…)

7. Line 92: Did fog vs. no fog have an impact on ns?

Assuming that "ns" refers to number concentrations: yes, fog or cloud immersion does affect the concentrations. We address this in detail in Supplement Sect. S4, where we dedicate an entire section to the changes in concentration during in-cloud (fog) periods. Please also refer to our answer to Referee 1, comment 1.

8. Line 101: "Laser-induced", not "light-induced".

The WIBS uses Xenon lamps, not Laser to excite particles, we therefore keep the expression as is.

9. Line 113: Please put these (A, B, AB etc.) into a table. It will be much easier to follow throughout the paper.

We put the particle types into a table and refer to it in line 115:

The particle types are listed in Tab. 1.

**Table 1.** Description of WIBS particle types and the channels they exhibit fluorescence in.

| Type | Active channels |
|------|-----------------|
| A | FL1 only |
| B | FL2 only |
| C | FL3 only |
| AB | FL1 and FL2 only |
| AC | FL1 and FL3 only |
| BC | FL2 and FL3 only |
| ABC | FL1, FL2 and FL3 |

10. Line 128: What kind of filter hold and filter system were used? What inlet was used (TSP? PM10?)? What was the pump and how was the orifice fabricated (unless it was commercial)?

Filter sampling was downstream the whole air inlet (equal to TSP), as stated in Fig. 1. In line 126 it is stated that sampling was done down the TSP inlet. Since in the rest of the manuscript we call the inlet "whole air inlet" we changed TSP to whole air inlet in line 126. In line 125 – 126 we write *"47 mm polycarbonat filters (Whatman®) with 0.2 µm pore size were installed in a metal filter housing down the TSP inlet (see Fig. 1) and changed once a week or more often."*

We change this to a more detailed description:

47 mm polycarbonate Nuclepore filters (111137, Whatman®) with 0.2 µm pore size were pre-cleaned with 10 % $H_2O_2$ and afterwards rinsed with deionized water that was passed through a 0.1 µm syringe filter (6784-2501, Whatman®). The filters were installed in a custom-made stainless steel filter holder.

The pump used was a scroll pump (SH-110, Agilent Technologies, Inc.) and the critical orifice was made out of a closed ISO-KF sealing, which had a small hole drilled through. The flow was measured with a flow meter (Defender 530+, Mesa Labs, Inc.). Please see the next comment for the incorporation of these details into the manuscript.

11. Line 128: What were the typical sampling times and volume collected?

In line 126, we originally stated: *"(...) and changed once a week or more often."* In line 128: *"The flow rate was held constant at 6.0 l min⁻¹ using a critical orifice and a pump."*

To improve clarity, we have removed the quoted part of the sentence in line 126 and added clarification after the sentence in line 128. The revised text now reads:

The standard flow rate was held constant at 6.0 l min⁻¹ using a critical orifice and a pump (SH-110, Agilent Technologies, Inc). The critical orifice was fabricated by drilling a small hole through a closed ISO-FK sealing. The flow rate was verified using a flowmeter (Defender 530+, Mesa Labs, Inc.). Sampling times were typically one week but ranged from 6 to 17 hours for certain filters at the beginning and end of the second part of the campaign. Consequently, the sampled air volume was approximately 60 m³ for most filters.

12. Line 130: What volume of water were the filters washed in?

We added "(5 or 8 ml)" after "Nanopure™ water" in line 130.

13. Line131: How many droplets? 96? 384?

Each INSEKT experiment comprises two 96-well PCR plates, totaling 192 wells. Of these, 32 wells were consistently filled with Nanopure™ water as background. The remaining wells were typically divided between two samples. The number of wells per sample varied depending on the number of dilutions used. For three filters (those with the shorter sampling durations) only the undiluted suspension (32 wells) and a 1:10 dilution (32 wells) was measured. For all other filters, either 80 or 160 wells were used to measure the undiluted suspension and multiple dilutions. We added the following sentence:

The aerosol suspension and dilutions are split into 50 µl aliquots and filled into PCR trays. Each INSEKT run consists of two 96-well plates (192 wells total), with 32 wells reserved for Nanopure™ water as backgounds. The remaining wells are divided between one or two samples, depending on the number of dilutions used.

14. Line 134: Please provide the INP equation, and in particular a brief description of the uncertainties that are mentioned and their calculation.

We revised the paragraph as follows:

The INP concentration in suspension $c_{INP,sus}$ is calculated from the frozen fraction according to Vali (1971):

$$c_{INP,sus} = -\frac{d}{V_{well}} \cdot \ln(f_l) \,, \tag{2}$$

where $d$ is the dilution scale, $V_{well}$ is the volume of one PCR well and $f_l$ is the fraction of liquid droplets as a function of the temperature $T$. To calculate the INP concentration per sampled air volume $c_{INP,air}$ , we use

$$c_{INP,air} = \frac{V_{sus}}{V_{air}} \cdot c_{INP,sus} \,, \tag{3}$$

where $V_{sus}$ is the volume of the whole suspension and $V_{air}$ is the volume of the sampled air. The uncertainties for the INP concentration follow the calculation by Agresti and Coull (1998) for a 95 % confidence interval. The uncertainty for the temperature is one standard deviation of the mean.

15. Line 139: In particular, such heat-labile INPs are believed to be "proteinaceous", since the heat is expected to denature proteins.

We changed "of biological origin" to "of proteinaceous origin".

16. Line 146: In which mode was PINE operated here? What was the temperature range? Was it operated at one temperature or was it thermocycled?

The PINE was operated on both temperature ramps and at constant temperature. One temperature ramp takes around 3 hours to cycle from -22 °C to -31 °C and back.
At line 146, we added:

During the PaCE22 campaign, the PINE was either operated in temperature ramp mode, or at constant temperature ranging between -22 °C and -31 °C.

17. Line 158: So there is data from every day that the trap was used?

The sentence we think Referee 2 refers to is "After sampling, the tape was cut into pieces representing one single day each." This sentence refers to the intended temporal resolution of the sampling setup. However, this does not imply that data was successfully collected for every day

of the campaign. Due to occasional interruptions or malfunctions of the Burkard trap, continuous daily coverage was not achieved. We clarify this point in the manuscript to avoid misunderstanding and changed the sentence to

After sampling, the tape was cut into pieces, each corresponding to a single day of sampling under normal operation. Due to occasional interruptions, however, data is not available for every day.

18. Line 170: Use "Environmental DNA (eDNA)…" the first time the technique is discussed (or in the sub-heading).

We use environmental DNA (eDNA) now in line 170.

19. Line 174: What is AIT?

We changed AIT to AIT Austrian Institute of Technology GmbH.

20. Section 2.6 (eDNA): Do any of these abbreviations mean anything or are they all proprietary? E.g. API, P3 AE, AIT? Define ITS, but also, what is meant by ITS2 region?

We added the following full forms, or explanations:

AP1 (proprietary Qiagen sample lysis buffer)

P3 (proprietary Qiagen precipitation buffer)

AE (proprietary Qiagen DNA elution buffer)

ITS (internal transcribed spacer, region between genes for ribosomal RNA)

ITS2-region (subregion in ITS between genes for 5.8S and 18S ribosomal RNA)

"LMP-Agarose" is replace by "Low Melting Point Agarose"

"Illumina MiSeq using V3 Chemistry" is replaced by "Illumina MiSeq using Illumina MiSeq Reagent Kit v3"

Illumina MiSeq is the sequencing machine and the Illumina MiSeq Reagent Kit v3 contains all the chemicals necessary to run the samples on the sequencer. Illumina is the company and MiSeq is one of their sequencers.

bp (basepairs) (building block in DNA – it is thus a length unit)

OTU (Operational Taxonomic Unit) (proxy for species in traditional biology)

"LGC" is replaced by "LGC Genomics GmbH" (LGC is a company name)

21. Section 2.7: For how long was the impactor usually run, and what volume of air was usually sampled? Was an after-filter installed to capture the smaller particles that passed through the main impactor cut-offs? What was the method for analysis under brightfield and fluorescence microscopy, what was being looked for?

We operated the impactor on a bidaily basis, between 6 and 20 h, with a mean sampling time of 12 h. The flow rate was 9 l/min. Therefore, the average collected volume was approximately 6.5

m$^3$. There was no after filter installed, as we were only interested in the stages with bigger cut-offs. In the main manuscript we only discuss qualitative analysis of the aluminum foils (we aimed to confirm the presence of fluorescent particles exhibiting morphological properties typical of fungal spores and that these particles make up the majority of particles > 1 µm aerodynamic diameter). Quantitative analysis of fluorescent particles (particle concentration) is discussed in the Supplement Sect. S8. To clarify, we have added a sentence after line 212, reading:

Sampling time was between 6 and 20 hours, with a mean collected volume of 6.5 m$^3$.

We extended the sentence in line 216-217 to:

The foils were qualitatively analyzed with a Nikon Eclipse Ci-L microscope (Nikon, Japan) under bright field and fluorescence setting ($\lambda_{ex}$=465-495 nm, $\lambda_{em}$=515-555 nm), to confirm the presence of fluorescent particles exhibiting morphological properties, typical of fungal spores. Quantitative analysis of fluorescent particles (particle concentration) is discussed in the Supplement Sect. S8.

22. Line 256: "since their high contribution during warmer months suggests an biological influence on their concentration." – But are these not already HFAPs? Unless I am missing something here, then there must naturally be a biological influence on the concentration of biological particles.

It is a common misconception that all fluorescent aerosol particles (FAPs), or particularly HFAPs, are biological particles. However, our results (and those of many other studies) indicate that this is not always the case. For example, we observe a strong linear correlation between B- and BC-type particle concentrations and elemental black carbon mass concentrations (see Fig. 5 (a) and Fig. S7), suggesting a non-biological source for at least some of these particles. This highlights the importance of carefully verifying the biological origin of specific particle types, and it is for this reason that we place particular emphasis on doing so in this study.

However, due to a comment by Referee 3, we changed the wording to

(…) since their high contribution during warmer months suggests a possible biological source.

23. Section 3.2: Could there be an influence of blowing snow on INP concentrations? (see for example, https://doi.org/10.1525/elementa.2024.00047). Are there any correlations with windspeed?

To analyze correlations between INPs measured with PINE and wind speed is beyond the scope of this paper. A detailed analysis of the PINE dataset will be presented in forthcoming research. However, we can provide insight into the relationship between HFAPs, particularly ABC particles, and wind speed in winter, as shown in Fig. 6 (c). To better visualize this relationship, we have adjusted the y-axis of Fig. 6 to a logarithmic scale. Additionally, we have expanded the discussion on potential winter sources of HFAPs.

However, the slight positive trend between ABC concentrations and wind speed suggest that the few potential local sources such as cellulolytic fungi, lichens, or mosses (see Sect. 3.2), may be more efficiently aerosolized under stronger winds (Ščevková et al., 2024; Armstrong, 1991; Kallio, 1970), transported to the site, or result from biological residues in blowing snow (Boetius et al., 2015).

Please also refer to our answers to Referee 1, comment 2.

24. Figure 2 caption: I would consider adding definitions of HFAPs and TAPs to this first figure caption where they are mentioned.

We agree and added the definitions.

25. Figure 3: Is there meant to be no temperature scale? This seems like an oversight.

We added the temperature axis, as depicted in the answers to Referee 1.

26. Figure 3: Even under maximum snow cover, HFAPs are higher than in early winter. Were there any potential sources of HFAPs in the area? Lichen on trees? Moss on rocks?

Although HFAPs in winter are predominantly composed of B and BC particles (see Fig. 2), which are likely non-biological in origin, as suggested by their strong correlation with elemental black carbon concentrations, we acknowledge that AB and ABC particles are still present during winter and the snow-covered period. In addition to the discussion of air masses arriving from the south, we now address potential local sources during snow-covered conditions beginning at line 285:

Potential local sources, even under snow-covered conditions, include cellulolytic fungi such as certain species within the order *Polyporales* which grow on trees and can produce fruitbodies that persist year-round. These fungi have been shown to sporulate under cold and harsh conditions (Gonthier et al., 2005). Aerosolized soredia from tree-dwelling lichens (Armstong et al., 1991) and spores from mosses (Ščevková et al., 2024), which grow on tree trunks or rocks may also contribute to PBAP emission. The fact that these organisms grow on trees, above the snowpack, may facilitate PBAP emissions even when the ground is covered in snow.

We come back to these potential sources in Sect. 3.4 and in the discussion about potential biological INPs other than fungi in Sect. 3.7. as suggested by Referee 4.

27. eDNA sequencing: It is unfortunate that this was not performed while the WIBS, Burkard or APS were not running. Can the authors comment on the inability to correlate this data with any of the other bioanalytical techniques and how that may impact the interpretation of the results?

We fully agree that the lack of parallel WIBS, Burkard, and APS data during the eDNA sampling period is unfortunate. This was not intentional but rather the result of an unfortunate coincidence: all three instruments experienced malfunctions during that specific time. Despite the absence of direct correlation with other techniques, the eDNA data still offers valuable insights. Nevertheless, the data was not collected simultaneously, which remains a big constraint, which is why we put more emphasis on mentioning this in the manuscript.

We added the following paragraph at line 437:

Unfortunately, due to simultaneous instrument malfunctions, no WIBS or Burkard measurements were available during the sequencing period. However, parallel impactor measurements and microscopic analysis indicate that the majority of particles with aerodynamic diameter > 1 µm during this time are likely fungal spores (see Fig. 8 for a representative example).

28. Line 287: I am surprised to see so little discussion of the HyIce INP results from the 2018 campaign, other than a very brief mention of Vogel et al. (2024) much later in the article.

We reference the HyICE-2018 campaign in line 288 in the context of snow cover (Schneider et al., (2021)). We again reference Schneider et al., (2021) in line 509, Vogel et al., (2024) in line 524 and Paramonov et al., (2020) in line 525.

29. The English is generally good throughout, but there are some notable spelling and grammatical errors throughout. The article should be proofread thoroughly prior to final submission.

We performed another thorough check of the text and point out that copernicus publications also provides outstanding typsetting and language copy-editing checks after acceptance.

30. Line 362: What are the sizes of these grains and how do they compare to the WIBS cut-off? I see this is discussed a little later and the pine pollen sizes were generally very large, but what about the other pollen grains? What is the detection limit of grain size on the Burkard trap analysis method? How do these other pollen grains, particularly those in the appropriate size range (if possible depending on the method's limitations) correlate with ABC_3?

According to the Paldat database (https://www.paldat.org/, accessed 19.06.2025), *Pinus* pollen ranges from 51 μm to over 100 μm, while cypress family pollen spans 10–100 μm. All other pollen types detected during the campaign fall within a range of approximately 10 μm to >100 μm. Beyond *Pinus*, four other pollen types show significant ($p < 0.05$) positive correlations with ABC_3 concentrations: *Betula* ($r = 0.50$, 10–25 μm), *Cyperaceae* ($r = 0.57$, 10–50 μm), *Picea* ($r = 0.31$, >100 μm), and unidentified pollen ($r = 0.41$).

Although particles >10 μm were only rarely detected as ABC_3 by the WIBS (see Fig. 7 (c), left), it is plausible that a small fraction of pollen grains within the detectable size range may contribute. However, since *Pinus* alone makes up 67 % of the total pollen and closely mirrors ABC_3 concentrations, the combined contribution of all other pollen types, especially those < 25 μm, cannot account for the observed ABC_3 particle levels in June.

Regarding the Burkard trap, its detection efficiency for small particles (<10 μm) is not well characterized. As noted in the manuscript (line 408 - 49), *"Furthermore, despite being used in aerobiological studies since more than 70 years, to the best knowledge of the authors, a characterization of the detection efficiency of a Hirst type spore trap has never been done for particles < 10 μm."* The device is primarily designed for larger particles such as pollen. Hirst (1952) reported an impaction efficiency of 62–95 % for *Lycopodium clavatum* (~35 μm), depending on wind speed.

We have extended the discussion as follows:

Despite the strong correlation between ABC_3 and pollen concentrations, it is unlikely that ABC_3 particles primarily represent pollen grains. Pine pollen, which accounts for 67 % of all counted grains, ranges from 51 to over 100 μm (Halbritter et al., 2021), well above the upper size limit reliably detected by WIBS. Moreover, the WIBS inlet system exhibits substantial losses for particles >10 μm (Gratzl et al., 2025), making the detection of pine pollen highly improbable. Even if a small subset of pine pollen were at or below 10 μm, WIBS would only capture a minor fraction, insufficient to explain the close match in concentrations. Other pollen types also show significant positive correlations ($p < 0.05$) with ABC_3 concentrations, including *Betula* (10–25 μm, $r = 0.50$), *Cyperaceae* (10–50 μm, $r = 0.57$), *Picea* (>100 μm, $r = 0.31$), and unidentified pollen types ($r = 0.41$; size data from https://www.paldat.org/, accessed 19.06.2025). While a small number of these smaller pollen grains, particularly *Betula*, may fall within the WIBS detection range and contribute to ABC_3 counts, their concentrations are too low to account for the total ABC_3 particle abundance observed.

31. Line 387: "These are starch granules which can be expelled from pollen grains…" – I do not think it is correct to definitively say that they are starch granules, unless I am mistaken pollen can expel various SPPs, including starch molecules, but also presumably pollen grains can expel a number of sup-pollen particles including various polysaccharides (of which starch is one type), as well as proteins and glycoproteins.

There are varying definitions of SPP in the literature. In this study, we follow the definition provided by Burkart et al. (2021), which characterizes SPP specifically as solid starch granules. All references to SPP in the manuscript are limited to particles that meet this definition. We acknowledge that pollen can also release other substances, however, these are typically soluble molecules rather than particulate matter and thus fall outside the scope of our discussion. Pollen fragments, on the other hand, are considered separately from SPP. We changed the sentence to

By the definition of Burkart et al., 2021, these are starch granules, (...)

32. Figure 7: What happens when ABC_3 are correlated against spores, while ABC* are correlated against pine pollen?

The correlation analysis gives the following results: ABC_3 to identifiable spores: r = -0.159 (p=0.200). ABC* against pine pollen: -0.27 (p= 0.026). To emphasize that they are not positively correlated, we added the following sentence at line 372:

Both the correlation between ABC* and pine pollen concentrations, and the correlation between ABC_3 and identifiable spore concentrations are negative, however, only the former is statistically significant (r = –0.23, p < 0.05).

33. Figure 7b: Although there's an apparent correlation, the number of data points above ~500 m-3 for pollen are very small.

We added the following to the sentence in line 365:

A scatter plot of ABC_3 versus pine pollen concentration (Fig. 7 (b, left)) shows a strong positive correlation (r = 0.92, p < 0.0001), although it is acknowledged that the number of data points above 400 m$^{-3}$ is limited.

34. Section 3.6: How does these results correlate with the ABC* discussion in the previous section? ABC is discussed here but not ABC*.

We assume Referee 2 refers to Sect. 3.7, as WIBS data is not discussed in Sect. 3.6. In this section, we focus on ABC concentrations rather than ABC* for two main reasons. First, the distinction between ABC* and ABC_3 particles is uncommon in existing literature. Thus, we aimed to report correlations consistent with previous studies. Second, ABC particles consist overwhelmingly of ABC* particles. As a result, the correlation analyses of INPs with ABC* and ABC particles yield nearly identical Pearson r values, with the greatest difference among significant results being ±0.01.

Nevertheless, we agree that this point warrants explicit mention. Therefore, we added the following sentence after line 489, following the statement: "*Since AB and ABC likely share a biological source, their combined correlation was tested but showed little difference from ABC alone.*" We appended:

Likewise, r-values for ABC* concentrations differ from those for ABC concentrations by a maximum of 0.01.

Furthermore, to clarify that primarily ABC concentrations are discussed in this chapter and not ABC* concentrations, and to address Referee 3, we added the following sentence at line 473:

We focus on ABC concentrations rather than ABC* or ABC_3, since ABC particles consist predominantly of ABC* particles and the distinction between ABC* and ABC_3 is uncommon in the existing literature, making comparisons with other studies more straightforward.

35. Line 455: This part is confusing; if 0.44 % of fungal reads are from Cladosporium then why is it being scaled to 100 %?

We thank Referee 2 for pointing out the potential confusion. To clarify, in the Burkard data, 100% of the identifiable fungal spores were *Cladosporium*, but we are not able to identify all fungal spores present in the air. The eDNA analysis, which provides a broader taxonomic overview, shows that *Cladosporium* represents only 0.44% of all fungal reads. Assuming that the eDNA data reflects the true relative abundance of fungal spores, we used the *Cladosporium* concentration from the Burkard data (30.5 m$^{-3}$) as representing 0.44% of the total fungal spores and scaled this up to estimate the total fungal spore concentration (6932 spores m$^{-3}$). We have revised the manuscript to explain this calculation more clearly as follows:

Although the two methods were not used simultaneously (with a two-day gap), we compare Burkard data from shortly before the DNA sampling period (September 11 to 13, 2023) with the three days of DNA sampling under similar, although colder weather conditions (September 16 to 18, 2023). During the relevant days, 100 % of identifiable fungal spores from the Burkard data were *Cladosporium* with a mean concentration of 30.5 m$^{-3}$. However, the Burkard method cannot identify all fungal spores to a taxonomic group, and it is unlikely that *Cladosporium* represents the entire fungal community. In contrast, eDNA analysis shows that *Cladosporium* accounts for only 0.44 % of all fungal reads. Assuming that the eDNA data reflects the true relative abundance of fungal taxa, the measured concentration of 30.5 m$^{-3}$ of *Cladosporium* spores corresponds to 0.44 % of the total fungal spore concentration. By scaling this proportion up to 100 %, we estimate the total fungal spore concentration to be approximately 6932 spores m$^{-3}$. This estimate is consistent, within one order of magnitude, with the concentration of fluorescent particles > 1 μm aerodynamic diameter (3010 m$^{-3}$) measured with the impactor and fluorescence microscope (see Supplement Sect. S8 for details).

36. Line 489: Bacteria may exhibit a range of activation temperatures, how can you be so sure that there is no influence of bacterial populations in the range of interest here (i.e. around -10 to -15 oC) and that it is nearly all fungal spores?

We thank Referee 2 for this important comment. We extended the discussion on bacteria. Please refer to our answer to comment 1.

37. Line 496: Do dust particles show up in any WIBS channels?

We thank Referee 2for this comment. Savage et al. (2017) conducted a laboratory study using a WIBS-4A with a 9σ fluorescence threshold and found that only a very small fraction of particles from 13 different dust types exhibited fluorescence. However, it is important to note that their results were obtained with a WIBS-4A and may not directly apply to measurements made with a WIBS-5/NEO, which might differ in detection sensitivity. Gao et al. (2024), using a WIBS-5/NEO and the same 9σ threshold, attributed B-, C-, and BC-type particles observed at Mount Helmos (Greece) to transported soil and mineral dust. This suggests that such dust may contribute to the HFAP signal under certain atmospheric conditions. To reflect this, we added the following to line 501 of the manuscript:

As HFAPs most likely consist of both biological and other particles (see Supplement Sect. S2), possibly also including soil dust and mineral dust (Gao et al., 2024), the significance of biological aerosols for INPs active at colder temperatures cannot be clarified.

38. Line 505: Qualitatively speaking, how do the heat treatments compare to the WIBS data in terms of HFAP loading?

The concentrations of heat-labile INPs active above −13.5 °C are several orders of magnitude lower than those of ABC and total HFAP particles. Even at an activation temperature of −18 °C, ABC and HFAP concentrations remain higher than those of heat-labile INPs. The smallest difference was observed between 24 and 31 May 2023, when ABC concentrations exceeded heat-labile INPs by a factor of 1.9. This further supports the conclusion that only a small fraction of ABC particles can account for the observed biological INP concentrations, as reflected by the scaling factor of 0.0022 used for ABC concentrations in Fig. 10a. At line 507 we added:

The concentrations of heat-labile INPs active above −13.5 °C are several orders of magnitude lower than those of ABC and total HFAP particles.

39. Line 540: Be a little careful here, several biological species defined as not having ice-nucleating properties have only been tested at warmer temperatures due to the type of instrument used or the nature of the method. Some may therefore exhibit activity at colder temperatures. I cannot speak to all of the examples provided here - some were clearly tested throughout a wide temperature range down to homogeneous freezing, so the statement of "Not all species within a genus exhibit ice nucleation" still holds true - but it is worth bearing in mind in general.

We agree that this is an important point made by Referee 2, and added a sentence at line 541, addressing this issue:

Not all species within a genus exhibit INA. Most tested species show very low or no INA in immersion freezing mode (including all tested species of *Aspergillus*, *Trichaptum*, and *Trichoderma*). It should be noted, however, that many species have only been tested at relatively high sub-zero temperatures due to methodological limitations and may still exhibit iNA at lower temperatures.

40. I am surprised to see so little discussion of the PINE data in Section 3.7 or 3.8 given that it was concurrent to and with similar resolution as the WIBS. It is certainly used (e.g. Figure 10b), but one might expect time series of PINE vs. WIBS for different groups (A, ABC) to help track the correlation. Is there are reason that such data is not shown?

In this study, we discuss INP data from INSEKT in detail, since high temperature biological INPs are the main focus here. PINE measured generally below -22 °C, a temperature, where biological aerosol is not expected to contribute significantly to the local INP population.

41. There are several similarities between this article and the authors' recent one in ESSD given that it was from the same campaign and discusses FBAPs and TAPs from the WIBS and other instruments. The authors should make clear in this article what is new compared to the ESSD publication.

We thank Referee 2 for raising this important point. Our data description paper in ESSD (Gratzl et al., 2024) presents and describes the WIBS dataset from the PaCE22 campaign in detail, including quality control procedures, but it does not include any supporting measurements. In contrast, the present manuscript addresses a different scientific objective: it investigates the biological origin of high temperature INPs in the European sub-Arctic by integrating WIBS data described in (Gratzl et al., 2024), four months of additional WIBS data and multiple complementary datasets, such as fungal spore counts, eDNA sequencing, and INP concentrations. We provide detailed analysis and interpretation of these combined observations.

We refer to the ESSD paper at several points of the manuscript for data reference and technical context and have clearly indicated in the data availability section (line 591) that Gratzl et al. (2024) is a data description paper ("*WIBS data from the PaCE22 campaign is available at the open data repository Zenodo under the doi 10.5281/zenodo.13885888 (Gratzl, 2024). A data description paper is provided by Gratzl et al. (2025)*"). We think that this contextualization makes the distinction between the two publications sufficiently clear for readers.

**Referee 3:**

This manuscript presents measurements in Northern Finland of INPs, HFAPs, spore and pollen counts, and DNA sequencing. The authors suggest that local fungal spores may dominate the high temperature INP population in the region, with possible implications for Arctic cloud processes under future climate scenarios. I believe this article is in scope with ACP after minor revisions based on the following comments.

Abstract:

1. L.10 It is great to use multiple data types (INP, HFAP and DNA), however please consider rephrasing "for the first time" to something like "for the first time to the best of our knowledge..." unless a thorough literature review confirms this claim.

We thank Referee 3 for this helpful comment. Although a thorough literature review was conducted, we have revised the wording to the suggested formulation, "for the first time, to the best of our knowledge," to ensure appropriate caution.

2. L.12 "Findings" is incorrectly capitalized, and the phrasing "Our findings indicate that" is used multiple times in close proximity (L.7 and L.12). Please revise for stylistic clarity and to avoid redundancy.

We changed "Findings" to lowercase, and changed the second "Our findings indicate that" in line 12 to "*The results suggest*".

3. L.13-14 it's unclear whether this study tested fungi for ice nucleation activity (INA), or whether this refers to results from previous studies. Please clarify and specify.

We thank Referee 3 for pointing out this lack of clarity. We changed the sentences to

eDNA analysis further reveals that airborne fungi are dominated by Basidiomycota, and that only a small fraction of the detected fungal genera has, to date, been tested for ice nucleation activity (INA) according to the literature. Among those reported in the literature, most exhibit very low or no INA.

Introduction:

4. L.20 Rather than citing only a review, please consider referencing original studies that include field-based INP concentration measurements to strengthen this statement.

We added the following references:

Vogel et al., 2024; Ladino et al., 2019; Si et al., 2018; Boose et al., 2016

5. L.33-34 The statement that biological INPs were discovered in the 1970s overlooks earlier work. Please revise or reformulate.

We thank Referee 3 for this comment. We changed the sentence to

Although biological ice nucleation was observed earlier, it was in the 1970s that bacterial cells were systematically shown to nucleate ice at relatively high sub-zero temperatures (Maki et al., 1974; Schnell and Vali, 1976; Maki and Willoughby, 1978).

6. L.51 "Less is known about the INA of fungal spores" could be more clearly phrased. Also, recent work from the group of Konrad Meister (e.g., Schwidetzky et al., 2023) should be referenced here to reflect the current state of knowledge.

We thank Referee 3 for this comment. We changed this paragraph to the following:

In contrast, less is known about the abundance and diversity of atmospheric ice nucleation active fungal spores. To date, only a small fraction of fungal species or genera have been tested for INA, and relatively few have been shown to nucleate ice at temperatures above –15 °C (Tarn et al., 2025; Fröhlich-Nowoisky et al., 2015; Haga et al., 2014, 2013; Morris et al., 2013; Pummer et al., 2013; Huffman et al., 2013; Pouleur et al., 1992; Jayaweera and Flanagan, 1982). Ascomycota (sac fungi) and Basidiomycota (club fungi) are the two major Phyla of fungi. Recent mechanistic insights by Schwidetzky et al. (2023) have demonstrated that INPs of the fungus *Fusarium* (Ascomycota) consist of small (~5.3 kDa) protein subunits that assemble into larger complexes. These findings emphasize that fungi can produce highly efficient biological INPs, although their prevalence in the atmosphere remains poorly understood.

7. L.55 It may strengthen this section to reference Sanchez-Marroquin et al. (2021), who combined INP measurements with SEM imaging and found mostly Ascomycota-type spores, with only occasional detection of basidiospores.

We extended the paragraph as follows:

Among those fungi tested for INA, most belong to the Ascomycota. However, a high abundance and diversity of Basidiomycota is expected in the atmosphere (Niu et al., 2024; Maki et al., 2023; Qu et al., 2021; Huffman et al., 2013). In the boreal forest which spans large areas of the sub-Arctic, Basidiomycota could dominate over Ascomycota (Qu et al., 2021; Sterkenburg et al., 2015). Nevertheless, Sanchez-Marroquin et al. (2021) found that most spores associated with INPs collected during aircraft campaigns over the southeast UK, resembled Ascomycota. Atmospheric fungal composition likely varies by region and ecosystem type, and findings from mid-latitude regions may not directly reflect the atmospheric fungal composition in boreal or sub-Arctic regions. Given the vast diversity of fungal species, many of which remain uncharacterized for INA, fungal spores, particularly from forest ecosystems, may represent a significant and underexplored source of atmospheric INPs.

L.60 Ideally define the cutoff of "high-temperature INPs" clearly (e.g., "active above –15 °C").

We added "(active above –15 °C)".

8. L.60 The current formulation implies that INPs active above –15 °C are specific to terrestrial sources in Arctic and sub-Arctic regions. Please rephrase or reflect the broader geographic relevance and support the statement with appropriate references e.g. Mason et al. 2015.

We acknowledge that Mason et al. (2015) concluded that terrestrial sources dominate over marine sources in terms of INP contributions. However, in this paragraph, we intentionally focused on studies conducted specifically in Arctic and sub-Arctic regions. Since the

measurements in Mason et al. (2015) were conducted on Vancouver Island, which is not considered part of the Arctic or sub-Arctic, we have chosen not to include it here. Nonetheless, we appreciate the point and have rephrased the paragraph for improved clarity. We also included Arctic haze in this paragraph, due to Referee 1's question. (The same revised paragraph is written under Referee 2's comment 4).

Recent studies have identified terrestrial environments as important, though not exclusive, sources of biological high temperature INPs (active above –15 °C) in the Arctic (Jensen et al., 2025; Wieber et al., 2025; Tobo et al., 2024; Pereira Freitas et al., 2023; Conen et al., 2016). In northern latitudes, INPs active above –15 °C show a distinct seasonal pattern, with concentrations peaking in summer during snow- and ice-free periods (Barry et al., 2025; Tobo et al., 2024; Pereira Freitas et al., 2023; Schneider et al., 2021; Wex et al., 2019). Fu et al. (2013) and Pereira Freitas et al. (2023) observed similar seasonal cycles in arabitol and mannitol, chemical tracers of fungal spores (Bauer et al., 2008), which they also attributed to terrestrial sources, in contrast to Arctic haze which dominates the aerosol composition during winter and early spring (Beck et al., 2024; Asmi et al., 2011; Quinn et al., 2007).

9. L.68 The connection between INP measurements and improved climate predictions is made too abruptly. Consider adding a more gradual buildup that explains how INPs influence cloud microphysics, which in turn affects cloud feedbacks and radiative forcing. This would help avoid overstatement.

We thank Referee 3 for this helpful suggestion. We agree that the transition to climate implications in the later paragraph was abrupt and could benefit from a more explicit connection to cloud microphysics. To address this, we have revised the paragraph in question to provide a more gradual buildup. It now reads:

These findings imply that high temperature INPs may become more prevalent in a warming climate, due to Arctic greening and prolonged snow free periods in the northern boreal forest (Barry et al., 2025; Berner et al., 2020; Tobo et al 2019.; Dankers and Christensen, 2005), and that fungal spores might contribute significantly to this highly active INP population. Since INPs influence cloud microphysics, particularly the phase, lifetime, and radiative properties of mixed-phase clouds (Bellouin et al., 2020), changes in their abundance or composition can have broader implications for Arctic cloud feedbacks and climate sensitivity. More data on the concentration and biological origin of high temperature INPs in northern latitudes could therefore enhance our understanding of cloud feedback mechanisms and potentially improve present and future climate predictions. Given the vast extent of the boreal forest, it should be considered a potentially critical source of biological INPs.

10. L.70 The hypothesis at the end comes across somewhat surprising. Please provide a rationale for this hypothesis earlier in the introduction.

We have added a sentence at the end of the paragraph discussed in our response to the previous comment, and we have also slightly softened the formulation of our hypothesis:

Given the vast extent of the boreal forest, it should be considered a potentially important source of biological INPs. In particular, fungal spores, which are abundant in boreal ecosystems, may represent a key fraction of these biologically derived INPs. Building on this, we hypothesize that locally emitted fungal spores from the northern boreal forest are a major source of high temperature INPs in the European sub-Arctic.

In addition, we have revised an earlier sentence in the introduction (line 58) to better support this rationale. The original sentence: *"Given the vast diversity of fungal species many of which remain uncharacterized for INA, their potential role in atmospheric ice nucleation remains largely unknown."* has been changed to:

Given the vast diversity of fungal species, many of which remain uncharacterized for INA, their potential contribution to atmospheric ice nucleation is still poorly understood. Fungal spores, particularly those from forest ecosystems, may represent a significant and underexplored source of atmospheric INPs.

The following paragraph is also revised in accordance with Referee 3's suggestions and incorporated an additional mention of Arctic haze, as recommended by Referee 2.

**To provide a clearer overview, we present the complete revised sections of the introduction starting from line 51.**

[revised manuscript text omitted]

Methods

11. L.93 Could the authors please specify how much the inlet is heated? Additionally, a brief discussion on how this may affect the collected DNA and INP concentrations would be appreciated.

The inlet construction is verified, and it fully complies with WMO/GAW (World Meteorological Organiszation/Global Atmosphere Watch Programm) and ACTRIS (Aerosol, Clouds and Trace gases Research Infrastructure) standards of an aerosol total inlet. More information on the inlet can be found in Komppula et al., (2005), where it is stated that *"The outer surface of the inlet is heated about 1°–2°C above zero to avoid the build up of ice and snow"* - however, the temperature of the inlet is not recorded. The mild heating of the inlet head is not expected to impact on the INP or DNA results, especially considering that the sampling room temperature is higher than the inlet temperature. We have amended the paragraph to:

The main inlet for aerosol instruments at the Sammaltunturi station is a whole air inlet with no cut-off diameter and thus collecting aerosols effectively also during fog and in-cloud periods. The inlet is ACTRIS (Aerosol, Clouds and Trace Gases Research Infrastructure) approved, and is located 2 m above the station's roof and approximately 6 m above ground. The inlet is slightly heated (to about 1 – 2 °C) to avoid snow and ice accumulation outside when the station is inside clouds or when it is snowing. The sampling into the aerosol instruments is conducted at room temperature. The aerosol instruments connected to the inlet have separate Permapure MD-700 nafion dryers for sample drying. More details about the inlet and sampling can be found in Backman (2025) and Komppula et al., (2005).

12. L.95 Are the WIBS concentrations reported per L or standard L?

The WIBS concentrations are reported per standard volume, as we state in line 95 – 96: *"All concentrations, except for the Burkard trap are reported for standard conditions (101325 Pa, 273.15 K)."*

13. L.120–121 The phrasing in this sentence could be refined for better readability.

We refined the sentence to:

Gratzl et al. (2025) also provide an overview of particle types associated with the different fluorescent categories, based on both laboratory and field studies.

14. L.159–162 Could the authors please clarify how plant spores were distinguished from fungal spores? Including typical examples or representative images would be helpful.

The identification of plant spores is made through morphological features. Plant spores such as *Equisetum*, fern spores, *Lycopodium* and trilete spores, are generally larger and have greater sphericity than fungal spores. To explain all these features would be out of scope for this manuscript. To clarify how we identified pollen, fungal spores and plant spores we extended the sentence in lines 159-160 to

Pollen grains and some plant and fungal spores were identified based on morphological features and counted from the samples using optical microscopy.

Furthermore, we added a microscopy picture of the Burkard tape from August 19, 2023, to the Supplement Sect. S6, where two trilete spores (which make up 87 % of identifiable plant spores) are visible. We extended the sentence in line 370 in the main manuscript to the following:

Identifiable plant spores were 87 % trilete spores, while the rest consists of *Equisetum*, fern and *Lycopodium*. A microscopy picture of a Burkard slide with two impacted trilete spores is shown in the Supplement Sect. S6.

We furthermore changed the heading of the Supplement Sect. S6 to

Microscopy pictures of fungal spores and trilete spores

and added the following sentence to line 119 in the Supplement:

Figure S12 shows a microscopic picture of two trilete spores and some fungal spores from August 19, at 1 pm (UTC+3).

[Figure]

**Figure S12**: An example of trilete spore counts from August 19, 2023, 1 pm (UTC+3). Black arrows point to reliable identified trilete spores. Some fungal spores are also visible but not marked.

15. Section 2.6 (DNA Sequencing) For clarity, it would be helpful to define the following abbreviations: AIT, AP1, P3, AE, LGC, ITS, LMP, V3, bp, OTU.

Please refer to our answer to Referee 2, comment 20.

Results and Discussion

16. L.257 Please clarify the meaning of "a biological influence on their concentration."

We acknowledge that the wording we used here is somewhat confusing. To clarify, we change it to

(...) since their high contribution during warmer months suggests a possible biological source.

17. L.329 Did the analysis focus only on the effect of rain on HFAP, or were INP, DNA, or spore counts also evaluated?

We thank Referee 3 for this comment. We only focused on the effect of rain on HFAP concentrations (and fractions). We now clarify this by writing the following at the end of a new paragraph, as described in the answer of Referee 1's first comment:

Further details on the effect of rain and cloud immersion on the concentrations and fractions of HFAPs and TAPs are discussed in Sect. S4 of the Supplement.

18. Supplement L.95 How frequently were rain events observed, and what was the approximate rainfall amount during those events? Could the authors also comment on how representative the two examples shown in Fig. S9 are?

We thank the referee for this valuable comment. To clarify, rain was recorded during approximately 23% of the total snow-free period, which is more than four months of observation. Rain events were defined as continuous precipitation periods, allowing for brief interruptions of up to 10 minutes. The duration of these events ranged from a few minutes to 95 hours, with a mean event duration of 170 minutes. Mean rain intensities per event varied considerably, from 0.06 mm/h to 10.4 mm/h, with an average intensity of 0.73 mm/h. However, due to the wide variability in event duration and intensity, mean values are not meaningful. We selected two representative periods shown in Fig. S9, encompassing a total of 53 distinct rain events. These range from brief light showers to prolonged episodes of heavy rainfall. We added the following paragraph to the Supplement Sect. S4.

Figure S9 also displays rain intensity for the selected periods. As discussed in Sect. 4 of the main manuscript, we generally did not observe an increase in HFAP concentrations during or after rainfall. Notably, during most rain events shown, the station was also immersed in cloud, suggesting that cloud and rain scavenging suppressed aerosol concentrations. This likely outweighs the rain-related enhancements in bioaerosols reported by previous studies (Rathnayake et al., 2017; Yue et al., 2016; Gosselin et al., 2016; Heo et al., 2014; Schumacher et al., 2013; Huffman et al., 2013; Prenni et al., 2013; Allitt et al., 2000; Hirst and Stedman, 1963; Gregory and Hirst, 1957). To provide context for the examples in Fig. S9, rain occurred during approximately 23 % of the snow free measurement period. Rain events varied greatly in duration and intensity, ranging from brief light showers to multi-day episodes of sustained precipitation. The two case studies shown represent different types of rainfall patterns commonly encountered during the campaign.

19. L.347–351 A brief interpretation of these results would be helpful to guide the reader through their implications.

We agree and revised this paragraph. Please refer to our answer to Referee 1, comment 1.

20. L.358 Is there a rationale for the selection of the FL1 threshold? Providing a brief justification would improve the transparency of the method.

we extended the sentence in line 357 - 358 to

We defined a new FL1 threshold at $1.0 \times 10^9$ intensity units, based on a local minimum of the trimodal intensity distribution (see Supplement Sect. S5, Fig. S10).

21. Fig. 7 (ABC vs identifiable spores) Maybe including logarithmic tick marks on the x-axis.

We added the tick marks.

22. L.395–396 Please clarify whether the following statements refer to ABC or ABC* particles. Additionally, rather than stating "marked orange on Fig. 7a," it may be clearer to say "marked as orange vertical lines in the lower subplot of Fig. 7a."

We thank Referee 3 for pointing this out. While we refer to ABC* concentrations in Fig. 7a, the statement also applies to ABC concentrations, as the ABC class is largely dominated by ABC* particles. We have revised the sentence accordingly to clarify this point:

One of high ABC* concentration (August 2, 2023), one of medium ABC* concentration (August 23, 2023) and one of low ABC* concentration (August 19, 2023) (marked as orange vertical lines in the lower subplot of Fig. 7(a)).

23. Table 1: The phrase "The concentration differences between identifiable spores and ABC particles are 1, 2 and 3 orders of magnitude" could be clarified.

We changed the phrase to

On August 2, 19, and 23, ABC concentrations exceed identifiable fungal spore concentrations by approximately one, two, and three orders of magnitude, respectively.

24. L.384 Please briefly explain the reasoning for ruling out ABC_3 particles as pollen grains here.

In response to Referee 2's comment, we have expanded the discussion on pollen grains and revised our wording to reflect a more nuanced interpretation, suggesting that it is unlikely that ABC_3 particles *primarily* consist of pollen grains. Please refer to our response to Referee 2, comment 30.

25. Section 3.8 It would be helpful to clarify whether you really mean ABC here (or rather ABC* or ABC_3 as differentiated in previous section).

We assume Referee 3 refers to Section 3.7, since WIBS data is not discussed in Section 3.8. Since Referee 2 made a similar point, we refer to our response to Referee 2, comment 34.

26. L.481 The statement that all INPs at –13.5 °C are ABC particles may be too strong. As correlation does not imply causation, a more cautious interpretation would be advisable.

We rephrased this sentence to

Thus, it is likely that the majority of INPs active at -13.5 °C in the immersion freezing mode are ABC particles, and therefore probably fungal spores.

27. L.492 Should the phrase "for temperatures > –8.5 °C" be revised to "for INPs active at temperatures above –8.5 °C"? Please clarify.

We rephrased as suggested.

28. L.519 This finding is consistent with Mignani et al. (2021), who observed that aerosol particles >2 µm were better predictors for INPs active at –15 °C than particles >0.5 µm at Weissfluhjoch, Switzerland.

We added the following statement at line 522:

Furthermore, Mignani et al. (2021) found that at Weissfluhjoch, Switzerland, INPs active at -15 °C were better predicted by considering only aerosol particles larger than 2 µm, rather than those larger than 0.5 µm.

**Referee 4:**

I agree with the overall assessment by the other three Referees that this is a well conceived and performed investigation worth publishing. My only comment in addition to what has already been said, relates to an issue triggered by Figure 3b in combination with Table 2. Figure 3b suggest that temperature alone can explain much of variation in ABC concentration, while snow cover does not seem to have an additional effect. Otherwise, there would be a step change between pink (snow covered period) and blue (no snow cover) dots in this graph. Yet, both clouds of dots nicely merge, which contradicts the statement in line 271 that "Snow cover significantly affects AB and ABC particle concentrations.." This observation makes me wonder whether the source of ABC particles is not to be sought above ground? Likely candidates are, like Referee #2 points out in their comment 26, lichen or moss on trees or rocks, which brings me to Table 2 from which the ice-nucleation active lichen mycobionts reported by Kieft and Ahmadjian (1989; https://doi.org/10.1017/S0024282989000599 ) are missing. It would be regrettable, if this potential source of ABC particles and INPs was not considered in a revised version of this paper.

We thank Referee 4 for this insightful comment. While we agree that Fig. 5 (b) gives the impression of a relatively smooth transition between the snow-covered and snow free periods, the influence of snow cover becomes much clearer when examining the temporal evolution in Fig. 3 (b). Additionally, the revised Fig. 6 (b), which shows the median ABC concentrations as a function of temperature across seasons, illustrates that during winter there is no clear monotonic increase in ABC concentrations with temperature.

The elevated ABC concentrations during the snow-covered period observed in Fig. 5 (b) can largely be attributed to higher concentrations during spring snowmelt, as seen in Fig. 3 (b). These events correspond to air masses arriving from the south, where snow had already melted, and are discussed in more detail in Sect. 3.2 of the main text and Sect. S3.1 of the Supplement.

To better illustrate that elevated ABC concentrations during the snow-covered period are primarily driven by such spring events, we revised Fig. 5 (b) to display the data by seasons rather than by snow-cover status. Nevertheless, we acknowledge that temperature is still a useful variable for describing the overall seasonal trend in concentrations. However, as shown in Fig. 6, temperature alone does not explain all the variability, as relative humidity and wind speed also contribute. Furthermore, Fig. 3 clearly shows a sharp drop in ABC concentrations coinciding with the onset of snow cover (particularly compared to TAPs), which is not evident in Fig. 5 (b) due to its lack of temporal resolution. We added this discussion into the figure caption:

[Figure]

**Figure 5**. Relation of concentration of particle types to meteorological parameters and air pollution. (a) Pearson correlation coefficient matrix of 1 hour mean values. Grey (empty) relations do not reach the 95 % confidence level. (b) scatter plot of the common logarithm of daily mean ABC particles against ambient temperature (r = 0.82). The transition between snow free and snow-covered periods (e.g., from autumn to winter) is less apparent than in Fig. 3 (b), as the temporal context is lost and ABC particles remain present to varying extents throughout the winter.

Nevertheless, we agree that snow cover does not explain all of ABC concentration variation and we acknowledge that we still find ABC particles during snow-covered periods without the influence of long range transport or snow melt south of Pallas. This indeed suggests that to some extend emissions from above ground might contribute to ABC (and INP) concentrations. We discuss moss and lichens as possible candidates in more detail and point out that a lot of tree decaying fungi, such as *Polyporales* (of which some have been detected by our eDNA analysis) also grow above ground on trees, year around. We now discuss these as candidates of ABC particles in winter (Sect. 2.3 and Sect. 2.4, see our answers to Referee 1, comment 2 and Referee 2, comments 23 and 26), and as possible INPs (see our answer to Referee 2, comment 1)

We thank the referee for the reference by Kieft and Ahmadjian (1989). However, in Tab. 2, we only list fungi that belong to a genus that we also detected via eDNA analysis. None of the described fungi/lichen in this reference was detected in our study.

However, we indeed overlooked one genus (*Rhizopus*) that is described in Jayaweera and Flanagan, (1982) and added it to table 2.

**Other changes:**

Equ. 1 should contain a "+" sign, rather than a "×" sign. We changed it accordingly.

**References:**

[revised manuscript text omitted]

---

## Author Response (AR2)

**Public justification (visible to the public if the article is accepted and published):**
I thank the authors for addressing the comments listed by the four referees; however, I invite you to address the following minor and technical comments before I can accept the manuscript for its publication.

We thank Referee 2 and the editor for their valuable feedback. Please see their comments in black, our response in blue and changes to the manuscript in red.

Reviewer #2 comments:

I am satisfied that the authors have responded appropriately to my queries and have updated the manuscript appropriately.

The only other question I have is whether the new Table 1 that shows the WIBS channels and types could be extended to include comments on the types of aerosol that each "type" might represent (including appropriate references that the authors have already largely provided in the main text), e.g. black carbon, fungal spores, bacteria, dust....

I understand that there may be many caveats to this and so the authors may choose to ignore it, but if possible it would be helpful to the reader. Aside from this minor comment, I am happy for the article to be published.

We thank Referee 2 for this suggestion. We have added aerosol classes that have been associated with the particle types to the table. We chose to include only studies that used the WIBS-5, WIBS-NEO, or WIBS-5/NEO instruments (which are identical except for the name) and excluded earlier models due to differences in detector gain settings. Additionally, we indicated the fluorescence threshold used in each study next to the corresponding references. We also added the following sentence in line 132:

The particle types are listed in Tab. 1 along with representative aerosol classes.

And deleted the sentence in line 137 – 138, as this is now covered in the caption of Tab.1

**Table 1.** Description of WIBS particle types and their corresponding fluorescence channels, along with representative particle classes identifies using a WIBS-5, WIBS-NEO or WIBS-5/NEO. Earlier WIBS models operated with different detector gain settings. A comprehensive overview covering all WIBS models is provided in Gratzl et al. (2025). (a): Katsivela et al. (2025), $9\sigma$ threshold; (b): Gratzl et al. (2025), 3 and $9\sigma$ threshold (c): Beck et al. (2024); $9\sigma$ threshold; (d) Gao et al. (2024), $9\sigma$ threshold; (e): Stone et al. (2021), threshold not reported; (f): Mampage et al. (2022); $3\times\overline{FT}$ threshold; (g): Hughes et al. (2020), $3\times\overline{FT}$ threshold; (h): Clancy et al. (2025), 6 and $9\sigma$ threshold; (i): Sarangi et al. (2022), threshold after Perring et al. (2015). SPP refers to sub-pollen particles.

| Type | Active channels | Representative aerosol classes |
|------|-----------------|--------------------------------|
| A | FL1 only | bacteria[a], microplastics[b] |
| B | FL2 only | eBC[c,d], pollen fragments/SPP[e,f,g] |
| C | FL3 only | fungal spores[h] |
| AB | FL1 and FL2 only | pollen fragments/SPP[f], microplastics[b] |
| AC | FL1 and FL3 only | fungal spores[i] |
| BC | FL2 and FL3 only | eBC[c,d], pollen[h], fungal spores[f,h], pollen fragments/SPP[e,f] |
| ABC | FL1, FL2 and FL3 | fungal spores[f,h,i], pollen[b], pollen fragments/SPP[f], microplastics[b] |

Editor comments:

Line 82: Replace "latitudes could therefor enhance" by "latitudes could therefore enhance"

Done!

Line 95 and along the document: Replace "565 meters above sea level, approximately 100 meters" by "565 m above sea level, approximately 100 m" for consistency.

Done! We also changed "meters" to "m" in the Supplement

Line 103: Replace "shown in In Fig.1 (b)." by "shown in Fig.1 (b)."

Done!

Lines 106, 119, 146, 188, 207, 250 and along the document: Replace "l min−1" by "L min−1"

Done!

Line 152 and along the document: Replace "ml" by "mL"

Done!

Lines 153, 210 (twice), 213, 214, 218, 291 and along the document: Replace "µl" by "µL"

Done!

Line 161: Something is wrong here: "concentration per sampled air volume CINP,air, we us"

We changed it to we use.

Lines 172, 558 (twice) and along the document: Replace "l" by "L"

We changed it, including the axis of Fig. 10.

Line 186 and along the document: Replace "3 meters" by "3 m" for consistency.

Done!

Line 251 and along the document: Replace "20 hours" by "20 h" for consistency.

Done!

Line 363: eBC was defined as equivalent black carbon in line 263.

The definition in line 263 is correct. We deleted de definition in line 363.

Line 432: Something is wrong here: "although it is acknowledged that the number of data points above 400"

We thank the editor for spotting this mistake. Something went wrong with the formatting. The missing part reads:

although it is acknowledged that the number of data points above 400 m$^{-3}$ is limited. Similarly, ABC_3 and total pollen concentration have an r-value of 0.90. The absolute concentrations of both ABC_3 and pine pollen are comparable (~45° linear fit).

Line 461: Replace "definition of (Burkart et al., 2021)," by "definition of Burkart et al. (2021),"

Done!

Line 534: The sentence is confusing: "the measured m−3 of Cladosporium spores corresponds"

We thank the editor. The correct sentence is

the measured concentration of 30.5 m$^{-3}$ of *Cladosporium* (...)

Line 546: Replace "relative humidity " by "R.H."

We now define R.H. as relative humidity in line 274 and then only use R.H. throughout the document.

Line 566: Replace "In (Fig. 10 (b)), the" by "In Fig. 10 (b), the"

Done!

Line 621: Replace "Furthermore, (Mignani et al., 2021) found" by "Furthermore, Mignani et al. (2021) found"

Done!

Figure 1: Replace "l min−1" by "L min−1"

Done!

Figure 3: How about if the color of the Snow depth y-axis is changed from black to light blue to improve the readability of the figure?

We thank the editor. We changed the axis to light blue.

Figure 5: Replace "(r = 0.82).The transition" by "(r = 0.82). The transition"

Done!

We furthermore deleted the reference to Brus et al., 2025 in line 101, as their preprint did not appear in time.